# Unpacking Reward Shaping: Understanding the Benefits of Reward Engineering on Sample Complexity

**Abhishek Gupta**[*]
University of Washington
abhgupta@cs.washington.edu

**Aldo Pacchiano**[*]
Microsoft Research, NYC
apacchiano@microsoft.com

**Yuexiang Zhai**
UC Berkeley, EECS
simonzhai@berkeley.edu

**Sham M. Kakade**
Harvard University
sham@seas.harvard.edu

**Sergey Levine**
UC Berkeley, EECS
svlevine@eecs.berkeley.edu

## Abstract

Reinforcement learning provides an automated framework for learning behaviors from high-level reward specifications, but in practice the choice of reward function can be crucial for good results – while in principle the reward only needs to specify what the task is, in reality practitioners often need to design more detailed rewards that provide the agent with some hints about how the task should be completed. The idea of this type of "reward-shaping" has been often discussed in the literature, and is often a critical part of practical applications, but there is relatively little formal characterization of how the choice of reward shaping can yield benefits in sample complexity. In this work, we build on the framework of novelty-based exploration to provide a simple scheme for incorporating shaped rewards into RL along with an analysis tool to show that particular choices of reward shaping provably improve sample efficiency. We characterize the class of problems where these gains are expected to be significant and show how this can be connected to practical algorithms in the literature. We confirm that these results hold in practice in an experimental evaluation, providing an insight into the mechanisms through which reward shaping can significantly improve the complexity of reinforcement learning while retaining asymptotic performance.

## 1 Introduction

Reinforcement learning (RL) in its most general form presents a very difficult optimization problem: when there are no constraints on the reward function or dynamics, a learning algorithm may need to exhaustively explore the entire state space to discover high-reward regions. Naïve algorithms that rely entirely on random exploration are known to be exponentially expensive [26, 3], and much of the theoretical work on efficient RL algorithms has focused on smarter exploration strategies that aim to more efficiently cover state space, typically in time polynomial in the state space cardinality [6, 8, 24, 55, 27, 32, 52]. Much of this work is based on upper confidence bound (UCB) principles and prescribes some kind of exploration bonus to prioritize exploration of rarely visited regions. Analogous strategies have also been employed in a number of practical RL algorithms [41, 40, 10,

---

[*]Equal Contributions

36th Conference on Neural Information Processing Systems (NeurIPS 2022).

12, 22, 36, 43, 42, 37]. However, perhaps surprisingly, much of the empirical work on reinforcement learning does not make use of explicit exploration bonuses or other dedicated exploration strategies, despite numerous theoretical results showing them to be essential to attain tractable sample complexity. Instead, practitioners often incorporate prior knowledge of each task into designing or shaping the reward function [31, 38, 39, 44, 5, 11, 50, 54], preferring this heuristic approach over the more principled exploration strategies. At this point, one may wonder why is reward shaping often practically preferable to dedicated exploration?

A likely answer to this question lies in the fact that even the best general-purpose exploration strategies still require visiting every state in the MDP at least once in the worst case. This is of course unavoidable without further assumptions. However, in practice, sample complexity that is polynomial in the size of the entire state space might *still* not be practical, and hence prior knowledge in the form of reward shaping is required to render such tasks tractable. Surprisingly, despite the widespread popularity of reward shaping in RL applications, the analysis of reward shaping has remained limited to proving policy invariance [31] or largely empirical observations, often relegating reward shaping to folk knowledge. In this work, we take a step towards studying the effect of reward shaping on the efficiency of RL algorithms, by asking the following question:

*Can we theoretically justify the sample complexity benefits that reward shaping from prior domain knowledge can provide for reinforcement learning?*

We aim to provide a set of tools that formally analyze how reward shaping can improve the complexity of tabula-rasa RL and better direct exploration. To perform this analysis, we first propose a simple modification to standard RL algorithms —"UCBVI-Shaped" that incorporates shaped rewards into optimism based exploration. We use this algorithm instantiation to then provide a regret analysis framework that studies how this introduction of shaped rewards can (in certain cases) provide much more directed optimism than uninformed exploration, while maintaining asymptotic performance.

To approach our analysis, we specifically consider problems where the reward shaping is provided through a term $\widetilde{V}$, a (potentially suboptimal) approximation of the optimal value function $V^\star$. In particular, we assume that the shaping $\widetilde{V}$ is a multiplicatively bounded approximation of the optimal value function $V^\star$, i.e., $\widetilde{V}(s) \leq V^\star(s) \leq \beta\widetilde{V}(s), \forall s$, for a finite multiplicative factor $\beta$. This type of shaped reward function $\widetilde{V}$ can be incorporated into a standard RL algorithm like UCBVI [8] through two channels: (1) bonus scaling – simply reweighting a standard, decaying count-based bonus $\frac{1}{\sqrt{N_h(s,a)}}$ by the per-state reward shaping and (2) value projection – adaptively projecting learned value functions into ranges of value functions derived from the reward shaping.

We show that this relatively simple algorithmic instantiation lends itself to an analysis that shows significant sample complexity benefits with shaping. Intuitively, the key pieces in our complexity analysis of UCBVI-Shaped are: (1) A multiplicative sandwich condition (via $\beta$) between $\widetilde{V}$ and $V^\star$ allows for the bonus scaling to depend on $\beta\widetilde{V}$ instead of a coarse approximation of $V^\star$ such as $H$. This allows for a reduction of complexity from a horizon $H$ dependent term to one scaling with $\beta\widetilde{V} \leq \beta V^\star$ by allowing for a faster decay of the exploration bonuses while still providing enough optimism. (2) A projection of the value function prevents "over optimism" by hastening the convergence of the empirical $\widehat{Q}$ functions during value iteration, thus allowing for faster detection of sub-optimal actions. This results in the ability to prune out large parts of the state space, as we also validate empirically.

To summarize, the key contribution of this work is to characterize how reward shaping can provably improve sample efficiency by providing gains in both $|\mathcal{S}|$ and $H$ dependent terms. We do so by analyzing the gains from reward shaping in two different ways: bonus scaling and value projection. We show that the "quality" (determined by $\beta$) of the provided shaping can significantly improve the sample efficiency of the resulting reinforcement learning algorithm, provide a set of analysis techniques to understand improvements in sample complexity from shaping, and confirm our findings with numerical experiments.

## 2   Related Work

**Regret Analysis in Finite Horizon Episodic Tabular MDPs.**   Recent research on regret analysis has studied both model-based and model-free RL methods. Model-based methods [23, 4, 8, 20, 17, 35] first learn a model from previous experience and use the learned model for future decision making. In contrast, model-free methods [24, 9, 55, 46, 30, 27] aim to learn the value function without the

model estimation stage and use the learned value function for decision making. Our analysis lies in the model-based framework and is similar to the setting of [8] but with additional assumptions on knowing $\widetilde{V}$ as a multiplicatively bounded approximation of $V^\star$. The main difference between our results and the aforementioned works is that we consider the reward shaping setting with a truncated interval assumption. As a result, we reduce the state dependency $|\mathcal{S}|$ to the some smaller "effective state space". Our work is also closely related to [21]. We discuss this in Section 3.

**Regret Analysis with Linear Function Approximation.** A recent line of literature has investigated the linear function approximation setting by assuming the transition kernels or the value functions can be represented by $d$-dimensional linear features [25, 53, 7, 56, 13, 48, 47, 45] or even general function classes [49]. With the aforementioned assumptions, the regret analysis will only depend on the ambient dimension $d$ (or other intrinsic complexity measure), instead of $S, A$ in the tabular setting [45, 48, 13, 56, 7, 52, 25], which could greatly decrease the complexity of learning. Instead of posing structural assumptions on the function class representing the MDP's values or dynamics, we ask the question of whether having approximate knowledge of $V^\star$ can improve the speed of learning.

**Practical Reward Design and Reward Shaping.** Ng et al. [31] proposed a potential-based shaping function $F$ that ensures policy invariance under transformation. However, unbiased potential based reward shaping is rarely used in practice. In most applications, heuristic reward design is carefully performed with potentially biased reward functions. In some large-scale practical RL tasks [11, 50, 51], the reward functions are heavily handcrafted based on prior knowledge. Besides reward engineering, a distinct line of work applies uninformed exploration algorithms like count-based reward shaping or intrinsic rewards to encourage greater state visitation [41, 40, 10, 12, 22, 36, 43, 42, 37, 28]. Importantly, these methods optimize for the worst case, as they try to cover *all* states since the exact location of the reward is unknown. In contrast, we look at the problem of incorporating domain knowledge via reward shaping into the exploration process. Closely related to our work is that of Cheng et al. [14], which studies how to incorporate shaping (heuristics) into the RL process via reducing the effective horizon. This work provides gains by reducing the horizon factor while our work provides gains by reducing the size of the effective state space that needs to be searched through.

## 3 Overview

We consider an episodic Markov Decision process $\mathcal{M} = (\mathcal{S}, \mathcal{A}, \mathbb{P}^\star, r, H)$ where $\mathcal{S}$ corresponds to the state space, $\mathcal{A}$ is the action space, $\mathbb{P}^\star$ is the transition operator, $r : \mathcal{S} \times \mathcal{A} \to [0,1]$ is the reward function and $H$ is the problem horizon. We use $|\mathcal{S}|, |\mathcal{A}|$ to denote the number of state and actions, and we use $\mathbb{P}^\star(\cdot|s,a)$ to denote the transition probability of state action pair $(s,a)$. The value function $V^\pi(s_0)$ of a policy $\pi$ starting at an initial state $s_0$ is defined as $V^\pi(s_0) := \mathbb{E}_\pi \left[ \sum_{h=0}^H r(s_h, a_h) \right]$, where $\mathbb{E}_\pi$ denote the transition dynamic of $\mathcal{M}$ under $\pi$. Similarly, $V^\star(s_0)$ is the value function of the optimal policy $\pi^\star$. We consider a sequential interaction between a learner and the MDP $\mathcal{M}$ occurring in rounds indexed by $t \in \mathbb{N}$. At the start of round $t$ the learner selects a policy $\pi_t$ that is used to gather a sample trajectory from $\mathcal{M}$. As is standard in the literature, we measure the learner's performance up to round $T$ by $\text{Regret}(T) := \sum_{t=1}^T V^\star(s_0) - V^{\pi_t}(s_0)$.

Our goal is to show that knowing a reward shaping term $\widetilde{V}$ allows for significantly more sample efficient learning, which with high probability has a sublinear regret upper bound. This bound has a leading term that depends on an "effective state space" of $\mathcal{M}$, determined by the quality of a reward shaping term $\widetilde{V}$ and the nature of the particular MDP being solved. This pruned effective state space can be much smaller than $|\mathcal{S}|$. Additionally, we will show an improved horizon dependence as the shaping term $\widetilde{V}$ allows for the bonus terms to be smaller and therefore decay faster thus replacing horizon factors of $H$ with $\beta\widetilde{V}$ ones.

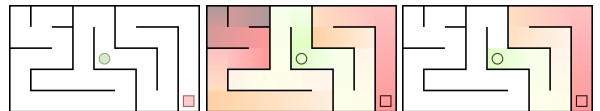

Figure 1: Reward shaping can allow for reduction of the "effective" state space size. In the above maze environment, the initial state is in the middle of the maze so finding the goal only requires solving half of the maze (**Left**). While uninformed state covering exploration (**Middle**) would go both directions since it has no knowledge of goal location, effective shaping (**Right**) would allow for halving of the effective state space. The green circle represents the starting point and the red square represents the goal. Different colors represents the landscape of the shaped value function (green indicates smaller value and red indicates larger value).

Intuitively, reward shaping allows for the consideration of a reduced effective state space. Consider the maze environment in Fig. 1 with agent starting in the middle. Without knowing where the goal is, an uninformed exploration algorithm needs to explore nearly the entire maze. However, an effectively incorporated reward shaping term $\widetilde{V}$ would allow the algorithm to prune out the entire left half of the maze, effectively halving the effective state space size. This can happen in two ways: firstly, if the shaping can directly indicate that certain actions are sub-optimal, then entire regions of the space can be eliminated from search. Secondly, even if states are not eliminated, if their corresponding bonuses are scaled according to their shaped value $\widetilde{V}$ from the shaping rather than uniformly with $H$, it limits unnecessary exploration of suboptimal states.

Based on this intuition, we propose modifications to a standard reinforcement algorithm that allows us to perform this analysis effectively. To aid our analysis, we introduce UCBVI-Shaped, a modification of the UCBVI algorithm [8] that uses an additional reward shaping term $\widetilde{V}$ in two ways: (1) Bonus scaling: $\widetilde{V}$ is used to provide an exploration bonus that combines inverse state visitation counts with reward shaping. This allows for the reduction of overoptimism and the dropping of the horizon $H$ dependence (2) Value projection: the shaping term is used to clip the empirical value function $\widehat{V}_h(s)$, which enables pruning unnecessary elements of the state space and allows for a bound that depends only on the effective state space size. Informally, our main result for UCBVI-Shaped can be summarized as follows:

**Theorem 3.1** (Main Informal). *With probability at least $1-\delta$, the regret of UCBVI - Shaped satisfies*[2]

$$\text{Regret}(T) = \mathcal{O}\left(B(\widetilde{V}, \mathcal{M})\log(T/\delta) + \text{Regret} - \text{UCBVI}(\mathcal{S}_{\text{remain}}, \mathcal{A}, T)\right).$$

*Here, $\mathcal{S}_{\text{remain}}$ corresponds to the set of states in $\mathcal{M}$ where the information contained in $\widetilde{V}$ was not enough to rule them out, and $\text{Regret} - \text{UCBVI}(\mathcal{S}_{\text{remain}}\mathcal{A}, T)$ is a UCBVI regret upper bound function evaluated on $\mathcal{S}_{\text{remain}}$ and $\mathcal{A}$. $B(\widetilde{V}, \mathcal{M})$ is a time-agnostic complexity measure that depends solely on the $\mathcal{M}$ and the quality of the shaping $\widetilde{V}$.*

We will define $\mathcal{S}_{\text{remain}}$ and $B(\widetilde{V}, \mathcal{M})$ and describe their relationship to $\widetilde{V}$ precisely in Section 5. If $\mathcal{S}_{\text{remain}} \ll \mathcal{S}$, the regret of UCBVI - Shaped can be substantially smaller than $\text{Regret} - \text{UCBVI}(\mathcal{S}, \mathcal{A})$, particularly since the first term of the regret grows logarithmically while the second scales with $\sqrt{T}$. We state our assumptions next:

**Assumption 1.** *The quality of the shaping term $\widetilde{V}$ is described by a parameter $\beta$. We assume access to a shaping "value function" estimators $\widetilde{V}_h : \mathcal{S} \to \mathbb{R}$ such that $V_h^\star(s) \leq \beta\widetilde{V}_h(s)$, for all $s \in \mathcal{S}$, $h \in [H]$ and for some $\beta \geq 1$.*

Instead of absorbing $\beta$ into the definition of $\widetilde{V}_h$ we allow $\widetilde{V}_h(s) < V_h^\star(s)$ for some states $s \in \mathcal{S}$. We'll show that learning a value of $\beta$ that turns this assumption true can be performed online. This assumption is intimately related to the optimistic $\widetilde{Q}$ assumptions of [21]. A thorough comparison with this work can be found in Appendix B.2.

**Assumption 2.** *We assume the reward functions satisfy $r(s,a) \in [0,1], \forall (s,a) \in \mathcal{S} \times \mathcal{A}$.*

**Assumption 3.** *The states $\mathcal{S}$ are $h-$indexed, i.e., the states reachable at time $h \in [H]$ are disjoint from the states reachable at time $h' \in [H]$ when $h \neq h'$.*

**Contributions.** Our main conceptual innovation is to introduce the notions of pseudosuboptimal and path-pseudosuboptimal states to quantify the "effective" size of the state space as a function of the quality of the shaping term $\widetilde{V}$ and use these notions to show how UCBVI-Shaped can attain significantly improved regret rates. In contrast with for example [21], where the regret rates will always scale at least with the number of states, our regret guarantees depend on an effective state size that may be orders of magnitude smaller. We believe this captures the real complexity improvement that reward shaping may yield, namely, avoid exploration of unnecessary areas of the state space. Other approaches such as [21] may (in general) at most yield an improvement in the dependence on the effective size of the action space. We further show that incorporating shaping into the exploration bonus term improves the horizon-dependence in the bound when the shaping is good enough, allowing us to replace a leading $H$ term with $\max_s \beta\widetilde{V}(s)$.

---

[2]Our main result, Theorem 5.2, is slightly more complex than this statement. We have chosen this simplified form to aid the reader to form the right intuition.

# 4 The UCBVI-Shaped Algorithm

To perform analysis of reward shaping, we build on the framework of the UCBVI algorithm [8]. UCBVI is an exploration algorithm based on the upper confidence bound, described in detail in Appendix D. This forms a convenient base algorithm for incorporating shaped rewards in a way that admits faster learning while maintaining asymptotic performance (as we will discuss in Section 5). Our algorithm, UCBVI-Shaped, is a combination of two changes to the upper confidence bound algorithm. First, we modify the bonus scaling to depend on $\widetilde{V}$. Second, we introduce a projection subroutine into value iteration, implemented as a standard clipping operation.

---

**Algorithm 1** UCBVI - Shaped

1: **Input** reward function $r$ (assumed to be known), confidence parameters
2: **for** $t = 1, ..., T$
3:     Compute $\widehat{P}_t$ using all previous empirical transition data as $\widehat{\mathbb{P}}_t(s'|s,a) := \frac{N_h^t(s,a,s')}{N_h^t(s,a)}, \; \forall h,s,a,s'$.
4:     Compute reward bonus $b_h^t(s,a)$ from Eqn. 1 (roughly of order $\frac{\widetilde{V}}{\sqrt{N(s,a)}}$)      ▷ Bonus scaling
5:     Run Value-Iteration with Projection (Algorithm 2).
6:     Set $\pi_t$ as the returned policy of VI.
7: **End for**

---

As in standard UCBVI, we define $N_h^t(s,a)$ to be the visitation count for the state-action pair $(s,a)$ at iteration $t-1$ for horizon $h$: $N_h^t(s,a) := \sum_{i=0}^{t-1} \mathbf{1}\left\{(s_h^i, a_h^i) = (s,a)\right\}$. Similarly to $N_h^t(s,a)$, we use $N_h^t(s,a,s') := \sum_{i=0}^{t-1} \mathbf{1}\left\{(s_h^i, a_h^i, s_{h+1}^i) = (s,a,s')\right\}$ as the visitation count of state-action pair $(s,a)$ and the subsequent state $s'$. We then use $\widehat{\mathbb{P}}_t(s'|s,a) := \frac{N_h^t(s,a,s')}{N_h^t(s,a)},^3 \; \forall h,s,a,s'$ to denote the empirical transition kernels at iteration $t$. UCBVI uses value iteration with the empirical transition function $\widehat{\mathbb{P}}_t$ and a reward function augmented with an exploration bonus, given by $r_h + b_h^t$. This is defined as a dynamic programming procedure that starts at $H$ and then proceeds backward in time to $h = 0$, updating according to Algorithm 1 and 2. The key algorithmic changes between UCBVI and UCBVI-Shaped are highlighted in blue: (1) scaling bonus $b_h^t$ by the shaping term $\widetilde{V}$ and (2) projecting the empirical value function $\widehat{V}_h^t(s)$ by an upper bound based on the sandwiched shaping, $\beta\widetilde{V}(s,a)$. As we discuss in Section 6, we also show how the sandwich factor $\beta$ does need to be provided beforehand but instead can be inferred through a straightforward technique for online model selection. While the approximate order of the bonus term is $\frac{\widetilde{V}}{\sqrt{N(s,a)}}$, a more detailed description can be found in Section 5.

---

**Algorithm 2** Value Iteration with Projection

1: **Input** $\left\{\widehat{P}_t, r + b_h^t\right\}_{h=0}^{H-1}$.
2: $\widehat{V}_H^t(s) = 0, \; \forall s, \widehat{Q}_h^t(s,a)$
3: **While** not converged
4:     $\widehat{Q}_h^t(s,a) = \min\left\{r_h(s,a) + b_h^t(s,a) + \widehat{\mathbb{P}}_t(\cdot|s,a) \cdot \widehat{V}_{h+1}^t, H\right\}$
5:     $\widehat{V}_h^t(s) = \min\left(\max_a \widehat{Q}_h^t(s,a), \beta\widetilde{V}(s)\right)$      ▷ Value projection
6:     $\pi_h^t(s) = \arg\max_a \widehat{Q}_h^t(s,a), \; \forall h,s,a.$

---

# 5 Analyzing UCBVI-Shaped

In this section, we will derive our main result on the sample complexity of the UCBVI-Shaped algorithm, and along the way introduce pseudosuboptimal and path-pseudosuboptimal states as a tool for deriving bounds that depend only on the "effective" state space as determined by the provided

---

$^3$The $\widehat{\mathbb{P}}_t$ here is dependent on the horizon $h$, but since we have assumed (Assumption 3) the states $s$ are $h$-indexed, we will use $\widehat{\mathbb{P}}_t$ for notation simplicity.

reward shaping. We will first introduce some notation to use in our analysis. As described in the previous section, UCBVI-Shaped proceeds in rounds. At the beginning of round $t$, the learner has access to an empirical model $\widehat{\mathbb{P}}_t$ built from the data collected up to iteration $t-1$. The bonus terms we consider are built by taking the empirical second moment of $\widetilde{V}$. This is related to the definition of $\text{bonus}_2$ in Azar et al. [8]. Importantly, the empirical value functions $\widehat{V}_h^t$ are clipped above by $\beta\widetilde{V}_h$:

$$b_h^t(s,a) = \min\left(16\beta\sqrt{\frac{\widehat{\mathbb{E}}_{s'\sim\widehat{\mathbb{P}}_t(\cdot|s,a)}[\widetilde{V}_{h+1}^2(s')|s,a]\ln\frac{2|\mathcal{S}||\mathcal{A}|}{\delta}}{N_h^t(s,a)}} + \frac{12\beta\widetilde{V}^{\max}}{N_h^t(s,a)}\ln\frac{2|\mathcal{S}||\mathcal{A}|t}{\delta}, 2\beta\widetilde{V}^{\max}\right), \quad (1)$$

where $\widetilde{V}_h^{\max} = \max_{s'}\widetilde{V}_h(s')$ and $\widetilde{V}^{\max} = \max_{s',h'}\widetilde{V}_{h'}(s')$.

Despite clipping and a modified bonus term, the $\widehat{Q}$ and $\widehat{V}$ values of UCBVI-Shaped are optimistic:

**Lemma 5.1.** *With probability at least $1-\delta$ we have*

$$\widehat{V}_0^t(s_0) \geq V_0^\star(s_0),\ \forall s_0 \in \mathcal{S}; \qquad and \qquad \widehat{Q}_h^t(s,a) \geq Q_h^\star(s,a), \quad \forall(s,a) \in \mathcal{S}\times\mathcal{A}, \quad (2)$$

*for all $t,h \in \mathbb{N}\times[H]$, where $\widehat{V}_h^t$ is computed via Algorithm 2.*

The proof of Lemma 5.1 can be found in Appendix A.1. Optimism (Lemma 5.1) and the simulation Lemma [2] imply that:

$$V^\star(s_0) - V^{\pi_t}(s_0) \leq \widehat{V}_1^t(s_0) - V^{\pi_t}(s_0) = \mathbb{E}_{\tau\sim\pi_t}\left[\sum_{h=1}^H b_h^t(s_h,a_h) + \left(\widehat{\mathbb{P}}_h^t(\cdot|s_h,a_h) - \mathbb{P}^\star(\cdot|s_h,a_h)\right)\cdot\widehat{V}_{h+1}^{\pi_t}\right].$$
$$(3)$$

We now turn our attention to characterize the information contained in $\widetilde{V}$.

### 5.1 Pruning of the State Space

Consider the following "surrogate" Q functions, $\widetilde{Q}_h^u : \mathcal{S}\times\mathcal{A}\to\mathbb{R}$ induced by $\widetilde{V}$ via:

$$\widetilde{Q}_h^u(s,a) = \mathbb{E}_{s'\sim\mathbb{P}(\cdot|s,a)}\left[r(s,a) + \beta\tilde{V}_{h+1}(s')\right].$$

By Assumption 1, we can bound $Q^\star$ via $Q_h^\star(s,a) \leq \widetilde{Q}_h^u(s,a)$. The basis of our main results is the following observation. If an action $a$ satisfies $\widetilde{Q}_h^u(s,a) < V_h^\star(s)$ for state $s$, then $a$ is a suboptimal action for state $s$. The projection (Step 5 of Algorithm 2) ensures the 'empirical' $Q-$functions $\widehat{Q}_h^t(s,a)$ of Algorithm 1 will quickly converge to values upper bounded by $\widetilde{Q}_h^u(s,a)$. Since optimism guarantees that $\widehat{Q}_h^t(s,\pi_*(s)) \geq V^\star(s)$, and the policy executed by UCBVI-Shaped is greedy w.r.t $\widehat{Q}_h^t$, actions belonging to state-action pairs such that $\widetilde{Q}_h^u(s,a) < V_h^\star(s)$ will quickly stopped being played by the algorithm. Moreover, all states that are only accessible through state-action pairs of this kind will also stop being visited by the algorithm after only a few iterations. In the subsequent discussion we will call

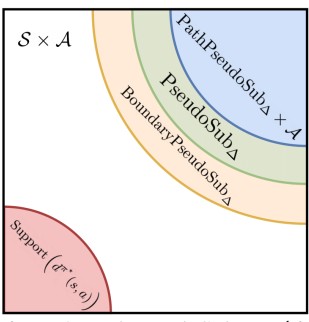

Figure 2: PathPseudoSub$_\Delta \times \mathcal{A}$ is split from Support($d^{\pi^*}(s,a)$) by BoundaryPseudoSub$_\Delta$. UCBVI-Shaped can avoid exploring over PathPseudoSub$_\Delta \times \mathcal{A}$.

PseudoSub to the set of state action pairs that are quickly 'pruned out' by UCBVI-Shaped, the set of states only accessible through these state action pairs as PathPseudoSub and the set of state-action pairs in PseudoSub that are not in PathPseudoSub $\times \mathcal{A}$ as BoundaryPseudoSub. Once all states in BoundaryPseudoSub have been visited enough, no states in PathPseudoSub will be visited again. Provided this happens sufficiently fast, we can show a regret bound that is independent on the size of PathPseudoSub. The subsequent discussion is aimed at formalizing this and culminates with our main result (Theorem 5.2).

Given a parameter $\Delta > 0$, we say that an action $a$ is $\Delta-$pseudosuboptimal[4] for state $s$ if $V_h^\star(s) \geq \Delta + \widetilde{Q}_h^u(s,a)$. We denote the set of $\Delta-$pseudosuboptimal state action pairs as:

$$\text{PseudoSub}_\Delta = \{(s,a) \in \mathcal{S}\times\mathcal{A}\text{ s.t. } V_h^\star(s) \geq \Delta + \widetilde{Q}_h^u(s,a)\}.$$

---

[4]We add the qualifier pseudo to our name to distinguish between suboptimality as captured by our surrogate $Q$ values and true suboptimality between the true values of $Q_h^\star$.

The intuition we want to capture is that all states $\tilde{s}$ that can be accessed only through traversing a state action pair in PseudoSub$_\Delta$, can be safely ignored because we can determine their suboptimality as soon as the state action pairs in PseudoSub$_\Delta$ that lead to $\tilde{s}$ are visited enough.

Now we define the set of $\Delta$-path-pseudosuboptimal states as the states that can be reached only by traversing a $\Delta-$pseudosuboptimal state action pair:

PathPseudoSub$_\Delta = \{s \in \mathcal{S}$ s.t. all feasible paths from initial states to $s$ intersect PseudoSub$_\Delta\}$.

Notice that there may be state action pairs in PathPseudoSub$_\Delta \times \mathcal{A}$ that may not be in PseudoSub$_\Delta$. In fact, there may exist states $s$ in PathPseudoSub$_\Delta$ such that no $(s,a)$ is in PseudoSub$_\Delta$ for all $a \in \mathcal{A}$. We will show that states in PathPseudoSub$_\Delta$ will not be explored by UCBVI-Shaped after a few iterations. Intuitively, this happens because the support of the state visitation distribution of the optimal policy does not contain any state in PathPseudoSub$_\Delta, \forall \Delta > 0$ (or equivalently, Support$(d^{\pi^\star}(s,a)) \cap$ PathPseudoSub$_\Delta = \emptyset, \forall \Delta > 0$). Hence, once the UCBVI-shaped identifies some sub-optimal state action pairs, the algorithm will not visit these state-action pairs again.

For any state action pair $(s,a)$, define its set of neighboring states Neighbor$(s,a)$ as the set of states with nonzero probability in $\mathbb{P}^\star(\cdot|s,a)$. By definition of PathPseudoSub$_\Delta$, for all $(s,a) \in (\mathcal{S} \times \mathcal{A})\backslash((\text{PathPseudoSub}_\Delta \times \mathcal{A}) \cup \text{PseudoSub}_\Delta)$, we have:

$$\text{Neighbor}(s,a) \subseteq \mathcal{S}\backslash\text{PathPseudoSub}_\Delta. \tag{4}$$

In other words, the neighborhood of any state action pair $(s,a)$ whose state is not in PathPseudoSub$_\Delta$ and such that $(s,a)$ is not $\Delta-$pseudo-suboptimal, are not in PathPseudoSub$_\Delta$. We also introduce the notion of "boundary pseudosuboptimal" state action pairs to capture the set of state action pairs that are $\Delta-$suboptimal but whose states are not in PathPseudoSub$_\Delta$.

BoundaryPseudoSub$_\Delta = \{(s,a) \in$ PseudoSub$_\Delta$ and $s \notin$ PathPseudoSub$_\Delta\}$.

Naturally, one of these states has to be traversed by any trajectory that contains any state in PathPseudoSub$_\Delta$.

Although we only use $\widetilde{Q}^u$ in the definition of PseudoSub$_\Delta$, PathPseudoSub$_\Delta$, and BoundaryPseudoSub$_\Delta$, the size of these sets is modulated by the scale of $\beta$ and the width of the intervals $\left[\widetilde{Q}^l(s,a), \widetilde{Q}^u(s,a)\right]$. We will show that (in the notation of Theorem 3.1) $\mathcal{S}_{\text{pruned}} \approx \mathcal{S}\backslash\text{PathPseudoSub}_\Delta$. With this notation, the formal version of our main results is stated as follows.

**Theorem 5.2.** *With probability at least* $1 - 6\delta$*, the regret of UCBVI-Shaped is upper bounded by*

$$\sum_{t=1}^{T} V^\star(s_0) - V^{\pi_t}(s_0) = \mathcal{O}\left(\min_\Delta\left(H\beta\widetilde{V}^{\max}\sqrt{|\mathcal{S}\backslash\text{PathPseudoSub}_\Delta||\mathcal{A}|T\ln\frac{\widetilde{V}^{\max}|\mathcal{S}||\mathcal{A}|T}{\delta}} + \right.\right.$$

$$\left.\left.\beta^2\left(\widetilde{V}^{\max}\right)^2 H^{1/2}|\text{BoundaryPseudoSub}_\Delta|^{1/2}\ln\frac{\widetilde{V}^{\max}|\mathcal{S}||\mathcal{A}|T}{\delta}\times\min\left(A(\Delta), B(\Delta)\right)\right)\right).$$

*for all* $T \in \mathbb{N}$*. Where* $A(\Delta) = \frac{|\mathcal{S}|^{1/2}|\mathcal{A}|^{1/2}}{\Delta}$ *and* $B(\Delta) = \frac{\beta\widetilde{V}^{\max}H^{1/2}|\text{BoundaryPseudoSub}_\Delta|^{1/2}}{\Delta^2}$.

Theorem 5.2 instantiates the desiderata of Theorem 3.1. Although this regret upper bound cannot be decomposed into a sum of two term as in Theorem 3.1. For any fixed $\Delta$, the regret upper bound has two components, one where $\mathcal{S}_{\text{remain}}$ can be identified with $|\mathcal{S}\backslash\text{PathPseudoSub}_\Delta|$ and a second one that scales logarithmically in $T$. In the next section we flesh out the steps in the proof of this result. The full argument can be found in Appendix A. The bonus scaling also allows us to ameliorate the horizon dependence of the upper bound. Instead of obtaining a $H^2$ dependence as the bound of theorem 7.1 in [2], the first term depends on $H\beta\widetilde{V}^{\max}$. Notice that the state dependence in the second term may be only be of order $\log(|\mathcal{S}|)$ if there is a $\Delta$ such that $|\text{BoundaryPseudoSub}_\Delta| \ll |\mathcal{S}|$ (albeit at the cost of a quadratic dependence on $1/\Delta^2$). Moreover notice that $|\text{BoundaryPseudoSub}_\Delta|$ could be much smaller than $|\text{PathPseudoSub}_\Delta|$ (the set of states that are reachable only by visiting states in $|\text{PseudoSub}_\Delta|$ ) thus showing that in some cases we can guarantee that even in the low order terms, the regret of UCBVI-Shaped that has polynomial dependence on an effective state space size that may be orders of magnitude smaller than the original one. The full version of this bound, with all the low order terms we have omitted for the sake of readability can be found in Appendix B.1, Theorem B.11. Note that Theorem 5.2 is a strict generalization to the UCBVI regret upper bounds, as setting $\Delta$ to a value that is smaller than the minimum gap recovers the exact result (Theorem 7.6in [2]).

## 5.2 Proof Intuitions and Sketch for Theorem 5.2

**Improved Horizon Dependence.** An empirical Bernstein bound shows that adding a bonus scaling (up to low order terms) with $\mathcal{O}\left(\sqrt{\widehat{\mathrm{Var}}_{s'\sim\widehat{\mathbb{P}}_h^t(\cdot|s,a)}(V_{h+1}^\star(s')|s,a)\ln\frac{2|\mathcal{S}||\mathcal{A}|t}{\delta}/N_h^t(s,a)}\right)$ is sufficient to ensure optimism. Since $V^\star$ is not known, this variance term can be substituted by the empirical second moment of $\beta\widetilde{V}_{h+1}$. Finally, the scaling of these terms can be upper bounded by $\beta\widetilde{V}^{\max}$. Without knowledge of $\widetilde{V}$, achieving this scaling would be challenging, since the only proxy for $V_{h+1}^\star$ available is $\widehat{V}_{h+1}^t$ or $H$ both of which may vastly overestimate it.

**State Pruning.** The value function clipping mechanism ensures that $\mathbb{E}_{s'\sim\widehat{\mathbb{P}}_h^t(\cdot|s,a)}\left[\widehat{V}_{h+1}^t(s')\right]\leq\beta\mathbb{E}_{s'\sim\widehat{\mathbb{P}}_h^t(\cdot|s,a)}\left[\widetilde{V}_{h+1}(s')\right]$ and therefore the empirical gap between $\widehat{Q}_h^t(s,a)$ and $V^\star(s)$ is decreasing at a rate of at most $\mathcal{O}\left(\beta\widetilde{V}^{\max}/\sqrt{N_h^t(s,a)}\right)$. Since optimism ensures that $\widehat{Q}_h^t(s,\pi^\star(a))\geq V^\star(s)$, once $\widehat{Q}(s,a)<V^\star(s)$, action $a$ will not be chosen anymore. Thus, for any $\Delta$, the number of times any state action pair in $\mathrm{PseudoSub}_\Delta$ may be visited by UCBVI-Shaped is upper bounded by $\beta^2\left(\widetilde{V}^{\max}\right)^2/\Delta^2$. Since any visit to a state action pair in $\mathrm{PathPseudoSub}_\Delta\times\mathcal{A}\cup\mathrm{PseudoSub}_\Delta$ requires a visit to a state in $\mathrm{BoundaryPseudoSub}_\Delta$, which allows us to bound the total number of visits to a state (or a trajectory containing such a state) in $\mathrm{PathPseudoSub}_\Delta\times\mathcal{A}\cup\mathrm{PseudoSub}_\Delta$ by $H|\mathrm{BoundaryPseudoSub}_\Delta|\beta^2\left(\widetilde{V}^{\max}\right)^2/\Delta^2$.

Finally, we can split (a version of) the regret decomposition in Eqn. 3 into two sums, one over state action pairs in $\mathcal{U}^{\mathrm{bad}}=\mathrm{PathPseudoSub}_\Delta\times\mathcal{A}\cup\mathrm{PseudoSub}_\Delta$ (or trajectories intersecting $\mathcal{U}^{\mathrm{bad}}$) and a second one over state action pairs in $\mathcal{U}^{\mathrm{good}}=(\mathcal{S}\times\mathcal{A})\setminus(\mathrm{PathPseudoSub}_\Delta\times\mathcal{A}\cup\mathrm{PseudoSub}_\Delta)$ (or trajectories without intersection with $\mathcal{U}^{\mathrm{bad}}$). We can then apply the upper bound on the visitation of state action pairs in $\mathrm{PathPseudoSub}_\Delta\times\mathcal{A}\cup\mathrm{PseudoSub}_\Delta$ to derive a regret upper bound over these states scaling with $1/\Delta$. Using the bound on the number of trajectories that intersect $\mathrm{PathPseudoSub}_\Delta\times\mathcal{A}\cup\mathrm{PseudoSub}_\Delta$, the trajectory decomposition yields the term scaling with $1/\Delta^2$ (but having only logarithmic state dependence) in Theorem 5.2. Recall that Eqn. 4 implies that the transition operators over state action pairs in $\mathcal{U}^{\mathrm{good}}$ have support only over $\mathcal{S}\setminus\mathrm{PathPseudoSub}_\Delta$. Our proofs use this fact to prove a polynomial dependence on $|\mathcal{S}\setminus\mathrm{PathPseudoSub}_\Delta|$ and not $|\mathcal{S}|$ in Theorem B.11 (see Appendix B.1), the full version of Theorem 5.2. Detailed proofs are in Appendix A.

## 6 Practical Considerations: Online Model Selection

Now one may notice that UCBVI-Shaped requires knowledge of the scaling $\beta$ is in order to actually perform the value projection. While this can be pre-provided by a user or set conservatively, in this section we discuss how this can be inferred by viewing this as an online model selection problem. In particular, given a set of $N$ different values of $\beta$ — $[\beta_1,\beta_2,\ldots,\beta_N]$, each of which parameterizes a different setting of the learning algorithm UCBVI-Shaped($\beta$), an online model selection algorithm such as Stochastic CORRAL [1, 34] or RegretBalancing [33, 15] jointly infers the value of $\beta$ and learns the appropriate value function online. In Appendix B.4 we show how these techniques can yield meaningful regret guarantees and we provide pseudocode.

## 7 Numerical Simulations

To show the practical relevance of our analysis on reward shaping we perform some numerical simulations on a family of maze environments with tabular state-action representations, as shown in Fig. 3. These environments have deterministic dynamics and reward. The reward is 0 at all states except a goal sink state, which has reward 1. These simulations are aimed at studying the following questions: **(1)** Does reward shaping improve sample complexity in these types of maze environments over uninformed UCBVI? **(2)** What is the relative importance of the bonus reweighting and the $\widetilde{V}$ projection? **(3)** How does the "suboptimality" of $\widetilde{V}$ impact the resulting sample complexity? **(4)** Does introducing decayed shaping actually allow for policy convergence? In these experiments, $\widetilde{V}$ is constructed by scaling the optimal value function $V^\star$ by per-state scaling factors sampled independently within the range $\beta$.

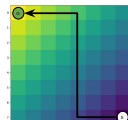 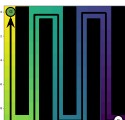 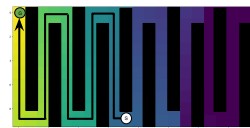

(a) Value function of Gridworld     (b) Value function of Single Corridor     (c) Value function of Double Corridor

Figure 3: Environments used for numerical simulations. **(Left)** Open Gridworld **(Middle)** Single corridor, agent starts bottom right has to reach a goal in top left **(Right)** Double corridor, agent starts in the middle and has to reach a goal on the left, with many irrelevant states on the right hand side

## 7.1 Does reward shaping help direct exploration over optimism under uncertainty?

We conducted numerical simulations on tabular environments to understand how reward shaping via UCBVI-Shaped provides benefits over standard UCBVI. Fig. 4 shows cumulative regret accumulated with different variants of UCBVI-Shaped (with both projection and bonus scaling), UCBVI-Shaped-P (with only projection), UCBVI-Shaped-BS (with only bonus scaling), UCBVI (standard UCBVI without shaping, as described in [8]). This is benchmarked across the three environments described in Fig. 3, with various levels of imperfect shaping applied by varying $\beta = \{1.5, 1.9\}$. As seen from Fig. 4, across all environments UCBVI-shaped with projection and bonus scaling performs most favorably, followed typically by UCBVI-Shaped-P, followed by UCBVI-Shaped-BS and UCBVI, suggesting that reward shaping can significantly help with learning efficiency.

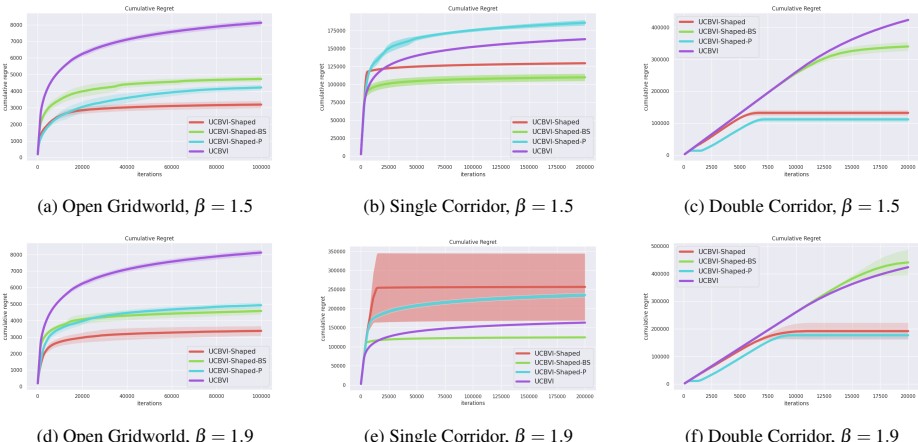

(a) Open Gridworld, $\beta = 1.5$     (b) Single Corridor, $\beta = 1.5$     (c) Double Corridor, $\beta = 1.5$

(d) Open Gridworld, $\beta = 1.9$     (e) Single Corridor, $\beta = 1.9$     (f) Double Corridor, $\beta = 1.9$

Figure 4: Cumulative regret for learning in various environments with varying amounts of shaping, as compared with UCBVI, and ablations UCBVI-Shaped-BS (no projection) and UCBVI-Shaped-P (no bonus scaling).

## 7.2 How does the effectiveness of reward shaping vary across environments?

We next conducted some numerical simulations across environments to understand how the nature of the environment itself affects the sample complexity of learning with shaping via UCBVI-Shaped. As shown in Fig. 6, we see that UCBVI shaped can be very effective in environments with many irrelevant sub-optimal paths like the

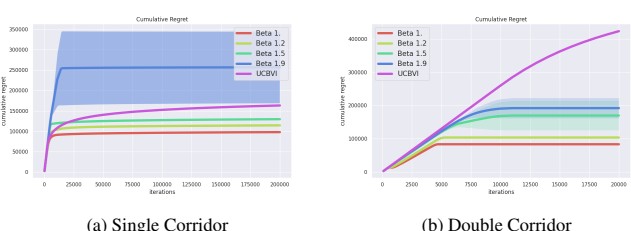

(a) Single Corridor     (b) Double Corridor

Figure 5: Effect of suboptimality of reward shaping on the performance of UCBVI-Shaped. While $\beta = 1.2, 1.5$ don't make much of a difference, very large $\beta$ leads to performance degradation

double corridor environment in Fig. 3, but is relatively less effective in environments where all exploration is directed the same way such as the single corridor. Even incorrect but optimistic shaping will provide guidance towards the goal, making UCBVI relatively less dominant in the single corridor environment as compared to the double corridor. This suggests that in environments where

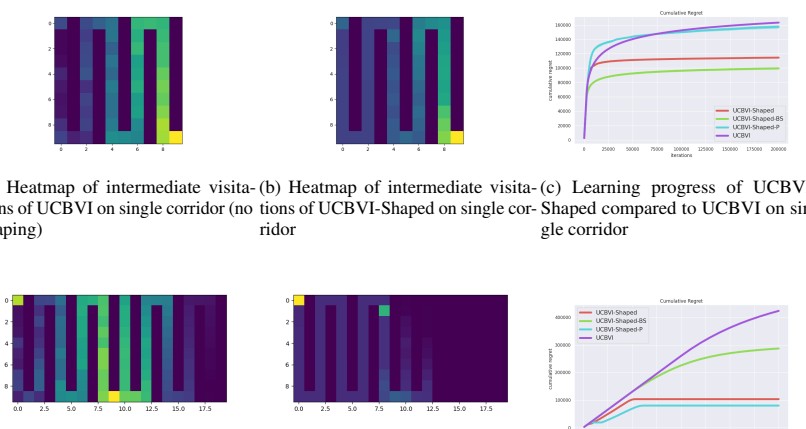

(a) Heatmap of intermediate visitations of UCBVI on single corridor (no shaping)

(b) Heatmap of intermediate visitations of UCBVI-Shaped on single corridor

(c) Learning progress of UCBVI-Shaped compared to UCBVI on single corridor

(d) Heatmap of intermediate visitations of UCBVI on double corridor (no shaping)

(e) Heatmap of intermediate visitations of UCBVI-Shaped on double corridor

(f) Learning progress of UCBVI-Shaped compared to UCBVI on double corridor

Figure 6: Visualization of how different environments are affected by reward shaping differently. (Left) intermediate visitations (Right) learning progress of UCBVI vs UCBVI-Shaped. The single corridor environment on the top sees much smaller gains for UCBVI-Shaped compared to double corridor environment.

ruling out an entire part of the exploration space is easy from the shaping, we can expect to see larger benefits.

### 7.3 How does the suboptimality of reward shaping affect learning?

We next compared how different levels of suboptimality of the $\beta$ sandwich term in the reward shaping affect cumulative regret across environments. As shown in Fig. 4 (a) and (d), we see that for environments with open paths (like the open gridworld), the shaping degradation has minimal negative effect until it gets very suboptimal. On the other hand, for corridor and double corridor (Fig. 5 (b)), where there are only a few paths to the goal, suboptimal reward shaping along those paths significantly hamper progress.

### 7.4 Is online UCBVI-Shaped able to infer $\beta$ online without prior knowledge?

As described in Section 6, UCBVI-shaped can be freed of the assumption of $\beta$ being known by performing online model selection of $\beta$ and learning values jointly. In particular, we use the Stochastic CORRAL algorithm [34], a variant of the method introduced in [1] to perform online model selection, with the episodic return being the requisite criterion for updating the model selection distribution. As we see in Fig 7, this scheme is able to show comparable results to when the actual $\beta$ is known beforehand, only degrading as the value of $\beta$ increases. This suggests that online UCBVI-shaped can be practical in regimes with moderate levels of value corruption.

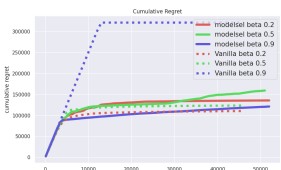

Figure 7: Understanding performance of online model selection in UCBVI-Shaped

## 8 Discussion

In this work, we take a step towards formally analyzing the benefits of reward shaping, proving that effective reward shaping can lead to more efficient learning than uninformed exploration strategies. In our analysis, we study an algorithm that incorporates reward shaping into a modified version of UCBVI, using it to modify bonuses and value function projection. Our analysis shows that incorporating shaped rewards allows for pruning significant parts of the state space and sharpening of optimism in a task directed way. This reduces the dependence of the regret bound on the state space size and on the horizon, depending on the quality of the shaping term and parameters of the MDP. This shows how reward shaping can direct exploration and provide significant sample complexity benefits while retaining asymptotic performance. We hope that this work is a step towards moving sample complexity analysis away from being reward agnostic to actually considering reward shaping more formally in analysis.

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
