$\text{PathPseudoSub}_\Delta, \forall \Delta > 0$ (or equivalently, $\text{Support}(d^{\pi^\star}(s,a)) \cap \text{PathPseudoSub}_\Delta = \emptyset, \forall \Delta > 0$). Hence, once the UCBVI-shaped identifies some sub-optimal state action pairs, the algorithm will not visit these state-action pairs again.

For any state action pair $(s,a)$, define its set of neighboring states $\text{Neighbor}(s,a)$ as the set of states with nonzero probability in $\mathbb{P}^\star(\cdot|s,a)$. By definition of $\text{PathPseudoSub}_\Delta$, for all $(s,a) \in (\mathcal{S} \times \mathcal{A}) \setminus ((\text{PathPseudoSub}_\Delta \times \mathcal{A}) \cup \text{PseudoSub}_\Delta)$, we have:

$$\text{Neighbor}(s,a) \subseteq \mathcal{S} \setminus \text{PathPseudoSub}_\Delta. \tag{4}$$

In other words, the neighborhood of any state action pair $(s,a)$ whose state is not in $\text{PathPseudoSub}_\Delta$ and such that $(s,a)$ is not $\Delta-$pseudo-suboptimal, are not in $\text{PathPseudoSub}_\Delta$. We also introduce the notion of "boundary pseudosuboptimal" state action pairs to capture the set of state action pairs that are $\Delta-$suboptimal but whose states are not in $\text{PathPseudoSub}_\Delta$.

$$\text{BoundaryPseudoSub}_\Delta = \{(s,a) \in \text{PseudoSub}_\Delta \text{ and } s \notin \text{

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

# A    Full Proofs

Let $f_h : \mathcal{S} \to \mathbb{R}$ be an arbitrary family of horizon indexed functions satisfying $\|f_h\|_\infty \le B$ for some $B > 0$. We assume $f_h$ is not a random variable as a function of $\mathcal{S}$. The following upper bounds hold,

**Lemma A.1.** *Fix $\delta \in (0,1)$, $\forall t \in \mathbb{N}$, $s \in \mathcal{S}$, $a \in \mathcal{A}$ and $h \in [H]$, with probability at least $1 - \delta$, we have*

$$\left| \widehat{\mathbb{P}}_t(\cdot|s,a)^\top f_{h+1} - \mathbb{P}^\star(\cdot|s,a)^\top f_{h+1} \right| \le 16 \sqrt{\frac{\mathrm{Var}(f_{h+1}(s')|s,a)\ln \frac{2|\mathcal{S}||\mathcal{A}|t}{\delta}}{N_h^t(s,a)}} + \frac{12B\frac{2|\mathcal{S}||\mathcal{A}|t}{\delta}}{N_h^t(s,a)}$$

$$\le 16B \sqrt{\frac{\ln \frac{2|\mathcal{S}||\mathcal{A}|t}{\delta}}{N_h^t(s,a)}} + \frac{12B\frac{2|\mathcal{S}||\mathcal{A}|t}{\delta}}{N_h^t(s,a)}$$

*for all $t \in \mathbb{N}$. Similarly with probability at least $1 - \delta$, we have*

$$\left| \widehat{\mathbb{P}}_t(\cdot|s,a)^\top f_{h+1} - \mathbb{P}^\star(\cdot|s,a)^\top f_{h+1} \right| \le 16 \sqrt{\frac{\widehat{\mathrm{Var}}(f_{h+1}(s')|s,a)\ln \frac{2|\mathcal{S}||\mathcal{A}|t}{\delta}}{N_h^t(s,a)}} + \frac{12B\frac{2|\mathcal{S}||\mathcal{A}|t}{\delta}}{N_h^t(s,a)}$$

*for all $t \in \mathbb{N}$. Where $\widehat{\mathrm{Var}}(f_{h+1}(s')|s,a) = \frac{1}{N_h^k(s,a)(N_h^k(s,a)-1)} \sum_{1 \le i < j < N_h^k(s,a)} (f_{h+1}(s_{h+1}^i) - f_{h+1}(s_{h+1}^j))^2.$*

*Proof.* Consider a fixed tuple $s,a,t,h \in \mathcal{S} \times \mathcal{A} \times \mathbb{N} \times [H]$. By the definition of $\widehat{\mathbb{P}}_t$, we have

$$\widehat{\mathbb{P}}_t(\cdot|s,a)^\top f_{h+1} = \frac{1}{N_h^t(s,a)} \sum_{i=1}^{t-1} \mathbf{1}\left\{ (s_{h'}^i, a_{h'}^i) = (s,a) \right\} f_{h+1}(s_{h'+1}^i).$$

Now denote $\mathcal{H}_{h,i}$ as the entire history from $t = 0$ to iteration $t = i$ where in iteration $i$, the history $\mathcal{H}_{h,i}$ includes all interactions from step 0 up to and including time step $h$. Next, $\forall i = 0, 1, \ldots, t-1$, define the random variables:

$$X_{i,h'}(s,a) = \mathbf{1}\left\{ (s_{h'}^i, a_{h'}^i) = (s,a) \right\} f_{h+1}(s_{h'+1}^i) - \mathbb{E}\left[ \mathbf{1}\left\{ (s_{h'}^i, a_{h'}^i) = (s,a) \right\} f_{h+1}(s_{h'+1}^i)|\mathcal{H}_{h,i} \right].$$

Notice that $\left| X_{i,h'} \right| \le B$, if $\mathbf{1}\left\{ (s_{h'}^i, a_{h'}^i) = (s,a) \right\} = 1$, else $\left| X_{i,h'} \right| = 0$ so that

$$\mathrm{Var}(X_{t,h}(s,a)|H_{h-1,t}) \le \begin{cases} 0 & \text{if } |X_{i,h'}| = 0 \\ B^2 & \text{o.w.} \end{cases}$$

An anytime Bernstein bound (see Lemma C.2) implies,

$$\left| \sum_{i=1}^{t-1} X_{i,h'}(s,a) \right| \le 16\sqrt{N_h^t(s,a)\mathrm{Var}(f_{h+1}(s')|s,a)\ln \frac{2|\mathcal{S}||\mathcal{A}|t}{\delta}} + 12B\frac{2|\mathcal{S}||\mathcal{A}|t}{\delta}$$

$$\le 16B\sqrt{N_h^t(s,a)\ln \frac{2|\mathcal{S}||\mathcal{A}|t}{\delta}} + 12B\frac{2|\mathcal{S}||\mathcal{A}|t}{\delta}.$$

With probability at least $1 - \delta$ for all $t \in \mathbb{N}$. Thus, we have

$$\left| \widehat{\mathbb{P}}_t(\cdot|s,a)^\top f_{h+1} - \mathbb{P}^\star(\cdot|s,a)^\top f_{h+1} \right| \le 16B\sqrt{\frac{\ln \frac{2|\mathcal{S}||\mathcal{A}|t}{\delta}}{N_h^t(s,a)}} + \frac{12B\frac{2|\mathcal{S}||\mathcal{A}|t}{\delta}}{N_h^t(s,a)}$$

with probability at least $1 - \delta$ for all $t \in \mathbb{N}$. The second inequality can be proven following the exact same proof process as described above, but instead making use of the empirical Bernstein bound of C.3 instead.

$\square$

The following bound relates the error of $\widehat{\mathbb{P}}_t$ and $\mathbb{P}^\star$.

**Corollary A.2** (State-action wise model error under $V^\star$ (Lemma 7.3 of [2])). *Fix $\delta \in (0,1)$, $\forall t \in [1,2,\dots T], s \in \mathcal{S}, a \in \mathcal{A}, h \in [H]$, consider $\forall V_h^\star : \mathcal{S} \to [0,H]$. With probability at least $1-\delta$, we have*

$$\left| \widehat{\mathbb{P}}_t(\cdot|s,a)^\top V_{h+1}^\star - \mathbb{P}^\star(\cdot|s,a)^\top V_{h+1}^\star \right| \leq$$

$$\min\left( 16\sqrt{\frac{\widehat{\mathrm{Var}}_{s'\sim\widehat{\mathbb{P}}_t(\cdot|s,a)}(V_{h+1}^\star(s')|s,a)\ln\frac{2|\mathcal{S}||\mathcal{A}|t}{\delta}}{N_h^t(s,a)}} + \frac{12V_{\max}^\star}{N_h^t(s,a)}\ln\frac{2|\mathcal{S}||\mathcal{A}|t}{\delta}, 2V_{\max}^\star \right),$$

*for all $t \in \mathbb{N}$. Where $\widehat{\mathrm{Var}}(V_{h+1}^\star(s')|s,a) = \frac{1}{N_h^k(s,a)(N_h^k(s,a)-1)}\sum_{1\leq i<j<N_h^k(s,a)}(V_{h+1}^\star(s_{h+1}^i)-V_{h+1}^\star(s_{h+1}^j))^2$ and $V_{\max}^\star = \max_s V^\star(s)$.*

*Proof.* A direct application of Lemma A.1 using $B = V_{\max}^\star$ yields the desired result. $\square$

A direct consequence of Lemma A.2 and Assumption 1 is that,

**Corollary A.3.** *The $V^\star$ "variance bonus" is upper bounded by the empirical $\widetilde{V}$ "second moment bonus":*

$$16\sqrt{\frac{\widehat{\mathrm{Var}}_{s'\sim\widehat{\mathbb{P}}_t(\cdot|s,a)}(V_{h+1}^\star(s')|s,a)\ln\frac{2|\mathcal{S}||\mathcal{A}|t}{\delta}}{N_h^t(s,a)}} \leq 16\beta \cdot \sqrt{\frac{\widehat{\mathbb{E}}_{s'\sim\widehat{\mathbb{P}}_t(\cdot|s,a)}[\widetilde{V}_{h+1}^2(s')|s,a]\ln\frac{2|\mathcal{S}||\mathcal{A}|}{\delta}}{N_h^t(s,a)}}$$

*and*

$$\frac{12V_{\max}^\star}{N_h^t(s,a)}\ln\frac{2|\mathcal{S}||\mathcal{A}|t}{\delta} \leq \frac{12\beta\widetilde{V}^{\max}}{N_h^t(s,a)}\ln\frac{2|\mathcal{S}||\mathcal{A}|t}{\delta}.$$

*Proof.* For any random variable $X \sim \mathbb{P}$,

$$\mathrm{Var}(X) \leq \mathbb{E}_{X\sim\mathbb{P}}[X^2].$$

Therefore, for all $s,a$ and any empirical distribution $\widehat{\mathbb{P}}_t(\cdot|s,a)$,

$$\widehat{\mathrm{Var}}_{s'\sim\widehat{\mathbb{P}}_t(\cdot|s,a)}(V_{h+1}^\star(s')|s,a) \leq \widehat{\mathbb{E}}_{s'\sim\widehat{\mathbb{P}}_t(\cdot|s,a)}\left[\left(V_{h+1}^\star(s')\right)^2|s,a\right].$$

Finally, by Assumption 1,

$$\widehat{\mathbb{E}}_{s'\sim\widehat{\mathbb{P}}_t(\cdot|s,a)}\left[\left(V_{h+1}^\star(s')\right)^2|s,a\right] \leq \beta^2\widehat{\mathbb{E}}_{s'\sim\widehat{\mathbb{P}}_t(\cdot|s,a)}\left[\widetilde{V}_{h+1}^2(s')|s,a\right].$$

The result follows. $\square$

Corollary A.3 and Lemma A.1 implies that with probability $1-\delta$ for all $t \in \mathbb{N}, s \in \mathcal{S}, a \in \mathcal{A}$ and $h \in [1,\cdots,H]$ the bonuses $b_h^t$ from Equation 1 satisfy,

$$\left|\left(\widehat{\mathbb{P}}_t(\cdot|s,a) - \mathbb{P}^\star(\cdot|s,a)\right)\cdot V_{h+1}^\star\right| \overset{(i)}{\leq} \min\left( 16\sqrt{\frac{\widehat{\mathrm{Var}}(V_{h+1}^\star(s')|s,a)\ln\frac{2|\mathcal{S}||\mathcal{A}|t}{\delta}}{N_h^t(s,a)}} + \frac{12V_{\max}^\star\ln\frac{2|\mathcal{S}||\mathcal{A}|t}{\delta}}{N_h^t(s,a)}, 2V_{\max}^\star \right)$$

$$\overset{(ii)}{\leq} b_h^t(s,a). \tag{5}$$

Where inequality $(i)$ follows by Corollary A.2 and $(ii)$ by Corollary A.3.

Define as $\widehat{V}_h^t$ to the stage $h$ value function of $\pi_t$ as computed in the learned model and using bonus augmented rewards (with bonuses defined as in 1) with the data collected up to round $t-1$. We start by showing $\widehat{V}_h^t$ is always larger than $V_h^\star(s)$.

We show the following supporting Lemma, a generalization of Lemma A.1 where the functions $f_h : \mathcal{S} \to \mathbb{R}$ satisfying $\|f_h\|_\infty$ are allowed to be random. This lemma is a slight refinement from its corresponding result in the literature, where the dependence on the size of the support of $|\mathbb{P}^\star(\cdot|s,a)|$ is upper bounded by $|\mathcal{S}|$. Having a more refined control on the size of the support of $\mathbb{P}^\star(\cdot|s,a)$ is what allows our final bounds to only depend on the size of the effective state space.

**Lemma A.4.** *Fix $\delta \in (0,1)$, $\forall t \in \mathbb{N}, s \in \mathcal{S}, a \in \mathcal{A}, h \in [H]$, with probability at least $1-\delta$, we have*

$$\left|\widehat{\mathbb{P}}_t(\cdot|s,a)^\top f_{h+1} - \mathbb{P}^\star(\cdot|s,a)^\top f_{h+1}\right|$$

$$\leq 16\beta \cdot \sqrt{\frac{|\text{support}(\mathbb{P}^\star(\cdot|s,a))|\widehat{\mathbb{E}}_{s'\sim\widehat{\mathbb{P}}_t(\cdot|s,a)}[\widetilde{V}_{h+1}^2(s')|s,a]\ln\frac{8B|\mathcal{S}||\mathcal{A}|t^3}{\delta}}{N_h^t(s,a)}}$$

$$+ \quad \frac{12B|\text{support}(\mathbb{P}^\star(\cdot|s,a))|}{N_h^t(s,a)}\ln\frac{8B|\mathcal{S}||\mathcal{A}|t^3}{\delta} + \frac{1}{t^2}$$

*simultaneously for all $\{f_h : \mathcal{S} \to \mathbb{R}\}_{h=0}^{H-1}$ such that $f_h(s) \in \left[\widetilde{V}_h, \beta\widetilde{V}_h\right]$ and $\|f_h\|_\infty \leq B$. Where* support$(\mathbb{P}^\star(\cdot|s,a))$ *corresponds to the size of the support of distribution* $\mathbb{P}^\star(\cdot|s,a)$.

*Proof.* The same proof template as in Lemma A.1 plus a union bound over a covering of the space of $f$ functions yields the desired result. Note that for all $s, a \in \mathcal{S} \times \mathcal{A}$,

$$\widehat{\text{Var}}(f_{h+1}(s')|s,a) \leq \widehat{\mathbb{E}}_{s'\sim\widehat{\mathbb{P}}_t(\cdot|s,a)}[f_{h+1}^2(s')|s,a] \leq \beta^2\widehat{\mathbb{E}}_{s'\sim\widehat{\mathbb{P}}_t(\cdot|s,a)}[\widetilde{V}_{h+1}^2(s')|s,a]$$

Therefore Lemma A.1 implies that for any fixed $f_{h+1}$, with probability at least $1-\delta'$

$$\left|\widehat{\mathbb{P}}_t(\cdot|s,a)^\top f_{h+1} - \mathbb{P}^\star(\cdot|s,a)^\top f_{h+1}\right| \tag{6}$$

$$\leq 16\beta \cdot \sqrt{\frac{\widehat{\text{Var}}_{s'\sim\widehat{\mathbb{P}}_t(\cdot|s,a)}(f_{h+1}(s')|s,a)\ln\frac{2|\mathcal{S}||\mathcal{A}|t}{\delta'}}{N_h^t(s,a)}} + \frac{12B}{N_h^t(s,a)}\ln\frac{2|\mathcal{S}||\mathcal{A}|t}{\delta'}$$

$$\leq 16\beta \cdot \sqrt{\frac{\widehat{\mathbb{E}}_{s'\sim\widehat{\mathbb{P}}_t(\cdot|s,a)}[\widetilde{V}_{h+1}^2(s')|s,a]\ln\frac{2|\mathcal{S}||\mathcal{A}|t}{\delta'}}{N_h^t(s,a)}} + \frac{12B}{N_h^t(s,a)}\ln\frac{2|\mathcal{S}||\mathcal{A}|t}{\delta'} \tag{7}$$

for all (fixed) $f \in \{f_h : \mathcal{S} \to \mathbb{R}\}_{h=0}^{H-1}$ and for all $s, a, h \in \mathcal{S} \times \mathcal{A} \times H$ simultaneously. We now apply a standard $\varepsilon$−net covering argument to the inequality we have proven above. Notice that regardless of the number of samples, support$(\widehat{\mathbb{P}}_t(\cdot|s,a)) \subseteq$ support$(\mathbb{P}^\star(\cdot|s,a))$. Thus, the only "entries" that matter in $f_{h+1}$ are those corresponding to states in support$(\mathbb{P}^\star(\cdot|s,a))$.

Let's consider an $\varepsilon$−net of $f_{h+1}$ restricted to support$(\mathbb{P}^\star(\cdot|s,a))$. Since for any $s \in \mathcal{S}$, we assume $f_{h+1} \in [\widetilde{V}_{h+1}, \beta\widetilde{V}_{h+1}]$, there exists an epsilon net $\mathcal{N}_\varepsilon$ under the infinity norm satisfying $|\mathcal{N}_\varepsilon| \leq \left(\frac{2B}{\varepsilon}\right)^{|\text{support}(\mathbb{P}^\star(\cdot|s,a))|}$. For any $f_{h+1}$ we denote by $f_{h+1}^\varepsilon$ its closest element in $\mathcal{N}_\varepsilon$ (in the infinity norm over support$(\mathbb{P}^\star(\cdot|s,a))$). The following holds,

$$\left|\left|\widehat{\mathbb{P}}_t(\cdot|s,a)^\top f_{h+1} - \mathbb{P}^\star(\cdot|s,a)^\top f_{h+1}\right| - \left|\widehat{\mathbb{P}}_t(\cdot|s,a)^\top f_{h+1}^\varepsilon - \mathbb{P}^\star(\cdot|s,a)^\top f_{h+1}^\varepsilon\right|\right| \leq$$

$$\left|\widehat{\mathbb{P}}_t(\cdot|s,a)^\top f_{h+1} - \widehat{\mathbb{P}}_t(\cdot|s,a)^\top f_{h+1}\right| + \left|\mathbb{P}^\star(\cdot|s,a)^\top f_{h+1}^\varepsilon - \mathbb{P}^\star(\cdot|s,a)^\top f_{h+1}^\varepsilon\right| \leq 2\varepsilon.$$

And therefore, setting $\delta' = \frac{\delta}{\left(\frac{2B}{\varepsilon}\right)^{|\text{support}(\mathbb{P}^\star(\cdot|s,a))|}} \leq \frac{\delta}{|\mathcal{N}_\varepsilon|}$, a union bound over all elements of $\mathcal{N}_\varepsilon$ implies that with probability at least $1-\delta$, we have

$$\left| \widehat{\mathbb{P}}_t(\cdot|s,a)^\top f_{h+1} - \mathbb{P}^\star(\cdot|s,a)^\top f_{h+1} \right|$$

$$\leq 16\beta \cdot \sqrt{\frac{|\mathrm{support}(\mathbb{P}^\star(\cdot|s,a))|\widehat{\mathbb{E}}_{s'\sim\widehat{\mathbb{P}}_t(\cdot|s,a)}[\tilde{V}_{h+1}^2(s')|s,a]\ln\frac{4B|\mathcal{S}||\mathcal{A}|t}{\varepsilon\delta}}{N_h^t(s,a)}}$$

$$+ \frac{12B|\mathrm{support}(\mathbb{P}^\star(\cdot|s,a))|}{N_h^t(s,a)}\ln\frac{4B|\mathcal{S}||\mathcal{A}|t}{\varepsilon\delta} + 2\varepsilon$$

for all $f_{h+1}$ simultaneously. Setting $\varepsilon = \frac{1}{2t^2}$ we get that with probability at least $1-\delta$,

$$\left| \widehat{\mathbb{P}}_t(\cdot|s,a)^\top f_{h+1} - \mathbb{P}^\star(\cdot|s,a)^\top f_{h+1} \right|$$

$$\leq 16\beta \cdot \sqrt{\frac{|\mathrm{support}(\mathbb{P}^\star(\cdot|s,a))|\widehat{\mathbb{E}}_{s'\sim\widehat{\mathbb{P}}_t(\cdot|s,a)}[\tilde{V}_{h+1}^2(s')|s,a]\ln\frac{8B|\mathcal{S}||\mathcal{A}|t^3}{\delta}}{N_h^t(s,a)}}$$

$$+ \frac{12B|\mathrm{support}(\mathbb{P}^\star(\cdot|s,a))|}{N_h^t(s,a)}\ln\frac{8B|\mathcal{S}||\mathcal{A}|t^3}{\delta} + \frac{1}{t^2}$$

for all $f_{h+1}$ simultaneously and for all $t \in \mathbb{N}$. This completes the result.

$\square$

## A.1 Proof of Lemma 5.1

In this section we show that optimism holds for all state-action pairs. We restate Lemma 5.1 for the reader's convenience.

**Lemma 5.1.** *With probability at least* $1-\delta$ *we have*

$$\widehat{V}_0^t(s_0) \geq V_0^\star(s_0), \ \forall s_0 \in \mathcal{S}; \qquad and \qquad \widehat{Q}_h^t(s,a) \geq Q_h^\star(s,a), \quad \forall(s,a) \in \mathcal{S} \times \mathcal{A}, \qquad (2)$$

*for all* $t,h \in \mathbb{N} \times [H]$, *where* $\widehat{V}_h^t$ *is computed via Algorithm 2.*

*Proof.* We prove via induction. At the additional time step $H$ we have $\widehat{V}_H^t(s) = V_H^\star(s) = 0$ for all $s$.

Starting at $h+1$, and assuming we have $\widehat{V}_{h+1}^t(s) \geq V_{h+1}^\star(s)$ for all $s$, we move to $h$ below.

Consider any $s,a \in \mathcal{S} \times \mathcal{A}$. First if $\widehat{Q}_h^t(s,a) = H$ then we have $\widehat{Q}_h^t(s,a) \geq Q_h^\star(s,a)$. The following inequalities hold,

$$\widehat{Q}_h^t(s,a) - Q_h^\star(s,a) = b_h^t(s,a) + \widehat{\mathbb{P}}_t(\cdot|s,a)\cdot\widehat{V}_{h+1}^t - \mathbb{P}^\star(\cdot|s,a)\cdot V_{h+1}^\star$$

$$\overset{(i)}{\geq} b_h^t(s,a) + \widehat{\mathbb{P}}_t(\cdot|s,a)\cdot V_{h+1}^\star - \mathbb{P}^\star(\cdot|s,a)\cdot V_{h+1}^\star$$

$$= b_h^t(s,a) + \left(\widehat{\mathbb{P}}_t(\cdot|s,a) - \mathbb{P}^\star(\cdot|s,a)\right)\cdot V_{h+1}^\star$$

$$\overset{(ii)}{\geq} b_h^t(s,a) - 16\sqrt{\frac{\widehat{\mathbb{E}}_{s'\sim\widehat{\mathbb{P}}_t(\cdot|s,a)}[\tilde{V}_{h+1}^2(s')|s,a]\ln\frac{2|\mathcal{S}||\mathcal{A}|t}{\delta}}{N_h^t(s,a)}} - \frac{12\widehat{V}^{\max}}{N_h^t(s,a)}\ln\frac{2|\mathcal{S}||\mathcal{A}|t}{\delta}$$

$$\overset{(iii)}{\geq} 0$$

Where $(i)$ holds because of the inductive optimism assumption. Inequality $(ii)$ holds because by Corollary A.2 we have,

$$\left(\widehat{\mathbb{P}}_t(\cdot|s,a) - \mathbb{P}^\star(\cdot|s,a)\right) \cdot V_{h+1}^\star \le 16\sqrt{\frac{\widehat{\mathrm{Var}}(V_{h+1}^\star(s')|s,a)\ln\frac{2|\mathcal{S}||\mathcal{A}|t}{\delta}}{N_h^t(s,a)}} + \frac{12V_{\max}^\star \frac{2|\mathcal{S}||\mathcal{A}|t}{\delta}}{N_h^t(s,a)}$$

and (*iii*) by Corollary A.3 applied to the definition of $b_h^t$.

$\square$

## B   Supporting Results for Section 5.1

A consequence of Lemma A.1, and Corollary A.3,

**Corollary B.1.** *Fix $\delta \in (0,1)$, $\forall t \in \mathbb{N}, s \in \mathcal{S}, a \in \mathcal{A}, h \in [H]$, consider $\forall \widetilde{V}_h : \mathcal{S} \mapsto [0,H]$ with probability at least $1 - \delta$, we have*

$$\left|\widehat{\mathbb{P}}_t(\cdot|s,a)^\top \widetilde{V}_{h+1} - \mathbb{P}^\star(\cdot|s,a)^\top \widetilde{V}_{h+1}\right| \le 16\sqrt{\frac{\widehat{\mathbb{E}}_{s'\sim\widehat{\mathbb{P}}_t(\cdot|s,a)}\left[\widetilde{V}_{h+1}^2(s')|s,a\right]\ln\frac{2|\mathcal{S}||\mathcal{A}|t}{\delta}}{N_h^t(s,a)}} + \frac{12\widetilde{V}^{\max}}{N_h^t(s,a)}\ln\frac{2|\mathcal{S}||\mathcal{A}|t}{\delta}.$$

We will now show the empirical $\widetilde{Q}$ values can be approximately upper bounded by the $\widetilde{Q}^u$ values.
Let's define the "tilde bonus" as $\tilde{b}_h^t(s,a) = 16\beta\sqrt{\frac{\widehat{\mathbb{E}}_{s'\sim\widehat{\mathbb{P}}_t(\cdot|s,a)}\left[\widetilde{V}_{h+1}^2(s')|s,a\right]\ln\frac{2|\mathcal{S}||\mathcal{A}|t}{\delta}}{N_h^t(s,a)}} + \frac{12\beta\widetilde{V}^{\max}}{N_h^t(s,a)}\ln\frac{2|\mathcal{S}||\mathcal{A}|t}{\delta}$.

**Corollary B.2.** *With probability at least $1 - 2\delta$ the empirical Q function of state action pair $(s,a)$ satisfies,*

$$\widehat{Q}_h^t(s,a) \le r(s,a) + b_h^t(s,a) + \tilde{b}_h^t(s,a) + \beta\mathbb{P}^\star(\cdot|s,a)^\top \widetilde{V}_{h+1} := \widetilde{Q}^u(s,a) + b_h^t(s,a) + \tilde{b}_h^t(s,a).$$

*for all $s,a \in \mathcal{S} \times \mathcal{A}$ and for all $h \in [0,\cdots,H-1]$ and all $t \in \mathbb{N}$.*

*Proof.* By definition $\widehat{Q}_h^t(s,a) = r(s,a) + b_h^t(s,a) + \widehat{\mathbb{P}}^t(\cdot|s,a)\widehat{V}_{h+1}^t$, since $\widehat{V}_{h+1}^t$ is clipped above by $\beta\widetilde{V}_{h+1}$,

$$\begin{aligned}
\widehat{Q}_h^t(s,a) &= r(s,a) + b_h^t(s,a) + \widehat{\mathbb{P}}^t(\cdot|s,a)\widehat{V}_{h+1}^t \\
&\le r(s,a) + b_h^t(s,a) + \beta\widehat{\mathbb{P}}^t(\cdot|s,a)\widetilde{V}_{h+1} \\
&\le r(s,a) + b_h^t(s,a) + \tilde{b}_h^t(s,a) + \beta\mathbb{P}^\star(\cdot|s,a)^\top \widetilde{V}_{h+1} \\
&= \widetilde{Q}^u(s,a) + b_h^t(s,a) + \tilde{b}_h^t(s,a).
\end{aligned}$$

The last inequality is a consequence of Corollary B.1. The result follows.

$\square$

Corollary B.2 implies that once $b_h^t(s,a) + \tilde{b}_h^t(s,a) < V^\star(s) - \widetilde{Q}^u(s,a)$ optimism guarantees that from that point on, action $a$ will never again be taken at state $s$. Indeed, since optimism (see Lemma 5.1) $\widehat{Q}_h^t(s,\pi_\star(s))$ is at least $V^\star(s)$, action $a$ will be dominated by action $\pi_\star(s)$ from that point onward. We make this intuition precise in the following Lemma by upper bounding the number of times action $a$ is taken at state $s$ when $(s,a) \in \mathrm{PseudoSub}_\Delta$.

**Lemma B.3.** *With probability at least $1 - 2\delta$ and for all $\Delta \in [0,1]$ the state action pairs $(s,a) \in \mathrm{PseudoSub}_\Delta$ satisfy the bound,*

$$N_{h(s)}^t(s,a) \le \frac{8192\beta^2 \times \left(\widetilde{V}^{\max}\right)^2 \cdot \ln\frac{2|\mathcal{S}||\mathcal{A}|t}{\delta}}{\Delta^2}$$

*For all $t \in \mathbb{N}$. Where $h(s)$ corresponds to horizon index of the state partitions that contains[5] state s.*

---

[5] Recall that by Assumption 3 the states are assumed to be $h-$indexed and therefore the state space can be written as $\mathcal{S} = \cup_{h\in[H]}\mathcal{S}_h$. This also implies that $N_h^t(s) = 0$ for all $h \ne h(s)$.

*Proof.* We start by assuming the events of Lemma 5.1 (optimism) and Corollary B.2 ($\widehat{Q}_h^t(s,a) \leq \widetilde{Q}^u(s,a) + b_h^t(s,a) + \tilde{b}_h^t(s,a)$) hold. We do not need any $\Delta$−dependent high probability event to hold for our results to be valid.

Once $b_{h(s)}^t(s,a) + \tilde{b}_{h(s)}^t(s,a) < \widetilde{V}(s) - \widetilde{Q}^u(s,a)$, action $a$ will never be taken again at state $s$. The following bound holds for $b_{h(s)}^t(s,a) + \tilde{b}_{h(s)}^t(s,a)$,

$$b_{h(s)}^t(s,a) + \tilde{b}_{h(s)}^t(s,a) \leq 32\beta \sqrt{\frac{\widehat{\mathbb{E}}_{s' \sim \widehat{\mathbb{P}}_t(\cdot|s,a)}[\widetilde{V}_{h(s)+1}^2(s')|s,a] \ln \frac{2|\mathcal{S}||\mathcal{A}|t}{\delta}}{N_{h(s)}^t(s,a)}} + \frac{24\beta \max_{s'} \widetilde{V}_{h(s)+1}(s')}{N_{h(s)}^t(s,a)} \ln \frac{2|\mathcal{S}||\mathcal{A}|t}{\delta}$$

$$\leq 32\beta \max_s \widetilde{V}_{h(s)+1}(s) \sqrt{\frac{\ln \frac{2|\mathcal{S}||\mathcal{A}|t}{\delta}}{N_{h(s)}^t(s,a)}} + \frac{24\beta \max_{s'} \widetilde{V}_{h(s)+1}(s')}{N_{h(s)}^t(s,a)} \ln \frac{2|\mathcal{S}||\mathcal{A}|t}{\delta}.$$

Assume $(s,a) \in \text{PseudoSub}_\Delta$. When

$$N_{h(s)}^t(s,a) > \frac{8192\beta^2 \times \max_{s',h'} \widetilde{V}_{h'}^2(s') \cdot \ln \frac{2|\mathcal{S}||\mathcal{A}|t}{\delta}}{\Delta^2}$$

we get

$$b_{h(s)}^t(s,a) + \tilde{b}_{h(s)}^t(s,a) < \Delta \leq V^\star(s) - \widetilde{Q}^u(s,a). \tag{8}$$

The result follows because as soon as $b_{h(s)}^t(s,a) + \tilde{b}_{h(s)}^t(s,a) < \Delta$ holds, the optimism guarantees that from that point on action $a$ will never again be taken at state $s$. Indeed, let $\pi^\star(s)$ be the optimal action at state $s$,

$$\widehat{Q}_{h(s)}^t(s, \pi^\star(s)) \geq Q^\star(s, \pi^\star(s)) = V^\star(s) \geq \widetilde{Q}^u(s,a) + \Delta$$

Where the last inequality holds because by definition $(s,a) \in \text{PseudoSub}_\Delta$.

Plugging in the bound of Corollary B.2, and invoking inequality 8,

$$\widehat{Q}_{h(s)}^t(s, \pi^\star(s)) \geq \widetilde{Q}^u(s,a) + \Delta > \widetilde{Q}^u(s,a) + b_{h(s)}^t(s,a) + \tilde{b}_{h(s)}^t(s,a) \geq \widehat{Q}_{h(s)}^t(s,a).$$

implying action $a$ will not be selected by the greedy policy induced by the empirical $Q$ function $\widehat{Q}_{h(s)}^t$. This finalizes the result. $\square$

**Lemma B.4** (Maximum Visitation of PseudoSuboptimal Pairs). *With probability at least $1 - 2\delta$, and for all $\Delta \in [0,1]$ and $T \in \mathbb{N}$ the number of episodes whose sample trajectories contain a state from* PathPseudoSub$_\Delta$ *is upper bounded by,*

$$\sum_{t=1}^T \mathbf{1}\left(\tau_t \cap (\text{PathPseudoSub}_\Delta \cup \text{PseudoSub}_\Delta) \neq \emptyset\right)$$

$$\leq |\text{BoundaryPseudoSub}_\Delta| \times \frac{8192\beta^2 \times \left(\widetilde{V}^{\max}\right)^2 \cdot \ln \frac{2|\mathcal{S}||\mathcal{A}|T}{\delta}}{\Delta^2}$$

*and*

$$\sum_{(s,a) \in (\{\text{PathPseudoSub}_\Delta \times \mathcal{A}\} \cup \text{PseudoSub}_\Delta)} N_{h(s)}^T(s,a)$$

$$\leq H \times |\text{BoundaryPseudoSub}_\Delta| \times \frac{8192\beta^2 \times \left(\widetilde{V}^{\max}\right)^2 \cdot \ln \frac{2|\mathcal{S}||\mathcal{A}|T}{\delta}}{\Delta^2}$$

*Where $h(s)$ corresponds to horizon index of the state partitions that contains state s.*

*Proof.* Let $\tau_t$ be the trajectory sampled by our algorithm at time $t$. Notice that whenever $\tau_t \cap$ PathPseudoSub$_\Delta \neq \emptyset$, there must be a state action pair in $\tau_t$ (occurring previous to the state in $\tau_t$ that lies in PathPseudoSub$_\Delta$) that is in BoundaryPseudoSub$_\Delta$, namely the first state-action pair belonging to PseudoSub$_\Delta$ in $\tau_t$. We obtain,

$$\sum_{t=1}^{T} \mathbf{1}\left(\tau_t \cap (\text{PathPseudoSub}_\Delta \cup \text{PseudoSub}_\Delta) \neq \emptyset\right) \leq \sum_{t=1}^{T} \mathbf{1}\left(\tau_t \cap \text{BoundaryPseudoSub}_\Delta \neq \emptyset\right).$$

Conditioning on the event from Lemma B.3 implies that for all $T \in \mathbb{N}$

$$\sum_{t=1}^{T} \mathbf{1}\left(\tau_t \cap \text{BoundaryPseudoSub}_\Delta \neq \emptyset\right) \leq \sum_{(s,a) \in \text{BoundaryPseudoSub}_\Delta} N_{h(s)}^{T}(s,a)$$

$$\leq |\text{BoundaryPseudoSub}_\Delta| \times \frac{8192 \beta^2 \times \max_{s',h'} \widetilde{V}_{h'}^2(s') \cdot \ln \frac{2|\mathcal{S}||\mathcal{A}|t}{\delta}}{\Delta^2}.$$

Thus,

$$\sum_{t=1}^{T} \mathbf{1}\left(\tau_t \cap (\text{PathPseudoSub}_\Delta \cup \text{PseudoSub}_\Delta) \neq \emptyset\right)$$

$$\leq |\text{BoundaryPseudoSub}_\Delta| \times \frac{8192 \beta^2 \times \max_{s',h'} \widetilde{V}_{h'}^2(s') \cdot \ln \frac{2|\mathcal{S}||\mathcal{A}|t}{\delta}}{\Delta^2}.$$

Finalizing the proof of the first bullet. To prove the second bullet, for any $t \in \mathbb{N}$ the following inequalities hold

$$\sum_{(s,a) \in (\text{PathPseudoSub}_\Delta \times \mathcal{A} \cup \text{PseudoSub}_\Delta)} N_{h(s)}^{t}(s,a)$$

$$\leq H \sum_{i=1}^{t} \mathbf{1}\left(\tau_t \cap (\text{PathPseudoSub}_\Delta \times \mathcal{A} \cup \text{PseudoSub}_\Delta) \neq \emptyset\right)$$

$$\leq H \sum_{i=1}^{t} \mathbf{1}\left(\tau_t \cap \text{BoundaryPseudoSub}_\Delta \neq \emptyset\right)$$

$$\leq H \sum_{(s,a) \in \text{BoundaryPseudoSub}_\Delta} N_{h(s)}^{t}(s,a)$$

$$\leq H \times |\text{BoundaryPseudoSub}_\Delta| \times \frac{8192 \beta^2 \times \max_{s',h'} \widetilde{V}_{h'}^2(s') \cdot \ln \frac{2|\mathcal{S}||\mathcal{A}|t}{\delta}}{\Delta^2}.$$

The result follows. $\qquad\square$

Let's now consider a refinement to the upper bound of Equation 3. For any[6] $T \in \mathbb{N}$,

$$\sum_{t=1}^{T} V^\star(s_0) - V^{\pi_t}(s_0) \leq \sum_{t=1}^{T} \mathbb{E}_{\tau \sim \pi_t} \left[ \sum_{h=1}^{H} b_h^t(s_h, a_h) + \left(\widehat{\mathbb{P}}_t(\cdot | s_h, a_h) - \mathbb{P}^\star(\cdot | s_h, a_h)\right) \cdot \widehat{V}_{h+1}^{\pi_t} \right]$$

$$\leq \underbrace{\sum_{t=1}^{T} \mathbb{E}_{\tau \sim \pi_t} \left[ \sum_{h=1}^{H} b_h^t(s_h, a_h) + \left(\widehat{\mathbb{P}}_t(\cdot | s_h, a_h) - \mathbb{P}^\star(\cdot | s_h, a_h)\right) \cdot \left(\widehat{V}_{h+1}^{\pi_t} - V_{h+1}^\star\right) \right]}_{\text{I}} +$$

$$\sum_{t=1}^{T} \mathbb{E}_{\tau \sim \pi_t} \left[ \sum_{h=1}^{H} \left(\widehat{\mathbb{P}}_t(\cdot | s_h, a_h) - \mathbb{P}^\star(\cdot | s_h, a_h)\right) \cdot V_{h+1}^\star \right]. \tag{9}$$

---

[6]From now on we'll use $T$ to index the final timestep of the regret sequence we are bounding.

By Equation 5 it follows that with probability at least $1 - \delta$ for all $s, a, h, T \in \mathcal{S} \times \mathcal{A} \times [H] \times \mathbb{N}$,

$$\sum_{t=1}^{T} \mathbb{E}_{\tau \sim \pi_t} \left[ \sum_{h=1}^{H} \left( \widehat{\mathbb{P}}_t(\cdot | s_h, a_h) - \mathbb{P}^{\star}(\cdot | s_h, a_h) \right) \cdot V_{h+1}^{\star} \right] \leq \sum_{t=1}^{T} \mathbb{E}_{\tau \sim \pi_t} \left[ \sum_{h=1}^{H} b_h^t(s_h, a_h) \right]$$

We will now focus on bounding I. We'll make use of Lemma 7.8 from [2]

**Lemma B.5** (Lemma 7.8 from [2]). *With probability at least $1 - \delta$ for all $t \in \mathbb{N}, s \in \mathcal{S}, a \in \mathcal{A}$, we have,*

$$\left| \widehat{\mathbb{P}}_t(s'|s,a) - \mathbb{P}^{\star}(s'|s,a) \right| \leq \sqrt{\frac{2\mathbb{P}^{\star}(s'|s,a) \ln\left(\frac{|\mathcal{S}||\mathcal{A}|tH}{\delta}\right)}{N_{h(s)}^t(s,a)}} + \frac{2\ln\left(\frac{|\mathcal{S}||\mathcal{A}|tH}{\delta}\right)}{N_{h(s)}^t(s,a)}$$

*Where $h(s)$ corresponds to horizon index of the state partitions that contains state $s$.*

A slight modification of the proof[7] of Lemma 7.7 in [2] yields,

**Lemma B.6** (Support Aware version of Lemma 7.7 in [2]). *With probability at least $1 - \delta$ for all $t \in \mathbb{N}, s \in \mathcal{S}, a \in \mathcal{A}$ and all $f : \mathcal{S} \to \mathbb{R}$ satisfying $f : \mathcal{S} \to [0, B]$ we have,*

$$\left| \left( \widehat{\mathbb{P}}_t(\cdot|s,a) - \mathbb{P}^{\star}(\cdot|s,a) \right) f \right| \leq \frac{\mathbb{E}_{s' \sim \mathbb{P}^{\star}(\cdot|s,a)} \left[ f(s') \right]}{H} + B \min \left( \frac{3 |\text{support}(\mathbb{P}^{\star}(\cdot|s,a))| H \ln\left(\frac{|\mathcal{S}||\mathcal{A}|tH}{\delta}\right)}{N_{h(s)}^t(s,a)}, 1 \right)$$

*Where $h(s)$ corresponds to horizon index of the state partitions that contains state $s$.*

---

[7]Instead of using $|\mathcal{S}|$ to uniformly bound the support of $\mathbb{P}^{\star}(\cdot|s,a)$ we explicitly write the bound in terms of its support

*Proof.* We start by conditioning on the event that Lemma B.5 holds. Take any function $f : \mathcal{S} \to [0,B]$.
We have,

$$\left|\left(\widehat{\mathbb{P}}_t(\cdot|s,a) - \mathbb{P}^\star(\cdot|s,a)\right)f\right| \leq \sum_{s'\in\mathcal{S}}\left|\left(\widehat{\mathbb{P}}_t(s'|s,a) - \mathbb{P}^\star(s'|s,a)\right)\right|f(s')$$

$$\leq \sum_{s'\in\mathcal{S}}\sqrt{\frac{2\mathbb{P}^\star(s'|s,a)\ln\left(\frac{|\mathcal{S}||\mathcal{A}|tH}{\delta}\right)f^2(s')}{N^t_{h(s)}(s,a)}} + \sum_{s'\in\mathcal{S}}\frac{2\ln\left(\frac{|\mathcal{S}||\mathcal{A}|tH}{\delta}\right)f(s')}{N^t_{h(s)}(s,a)}$$

$$\overset{(i)}{\leq} \sum_{s'\in\mathcal{S}}\sqrt{\frac{2\mathbb{P}^\star(s'|s,a)\ln\left(\frac{|\mathcal{S}||\mathcal{A}|tH}{\delta}\right)f^2(s')}{N^t_{h(s)}(s,a)}} + \frac{2B|\text{support}(\mathbb{P}^\star(\cdot|s,a))|\ln\left(\frac{|\mathcal{S}||\mathcal{A}|tH}{\delta}\right)}{N^t_{h(s)}(s,a)}$$

$$\overset{(ii)}{\leq} \sqrt{|\text{support}(\mathbb{P}^\star(\cdot|s,a))|}\sqrt{\frac{\sum_{s'\in\mathcal{S}}2\mathbb{P}^\star(s'|s,a)\ln\left(\frac{|\mathcal{S}||\mathcal{A}|tH}{\delta}\right)f^2(s')}{N^t_{h(s)}(s,a)}} +$$

$$\frac{2B|\text{support}(\mathbb{P}^\star(\cdot|s,a))|\ln\left(\frac{|\mathcal{S}||\mathcal{A}|tH}{\delta}\right)}{N^t_{h(s)}(s,a)}$$

$$= \sqrt{\frac{2|\text{support}(\mathbb{P}^\star(\cdot|s,a))|BH\ln\left(\frac{|\mathcal{S}||\mathcal{A}|tH}{\delta}\right)}{N^t_{h(s)}(s,a)}\frac{\sum_{s'\in\mathcal{S}}\mathbb{P}^\star(s'|s,a)f^2(s')}{BH}} +$$

$$\frac{2B|\text{support}(\mathbb{P}^\star(\cdot|s,a))|\ln\left(\frac{|\mathcal{S}||\mathcal{A}|tH}{\delta}\right)}{N^t_{h(s)}(s,a)}$$

$$\overset{(iii)}{\leq} \frac{|\text{support}(\mathbb{P}^\star(\cdot|s,a))|BH\ln\left(\frac{|\mathcal{S}||\mathcal{A}|tH}{\delta}\right)}{N^t_{h(s)}(s,a)} + \frac{\sum_{s'\in\mathcal{S}}\mathbb{P}^\star(s'|s,a)f^2(s')}{BH} +$$

$$\frac{2B|\text{support}(\mathbb{P}^\star(\cdot|s,a))|\ln\left(\frac{|\mathcal{S}||\mathcal{A}|tH}{\delta}\right)}{N^t_{h(s)}(s,a)}$$

$$\overset{(iv)}{\leq} \frac{|\text{support}(\mathbb{P}^\star(\cdot|s,a))|BH\ln\left(\frac{|\mathcal{S}||\mathcal{A}|tH}{\delta}\right)}{N^t_{h(s)}(s,a)} + \frac{\sum_{s'\in\mathcal{S}}\mathbb{P}^\star(s'|s,a)f(s')}{H} +$$

$$\frac{2B|\text{support}(\mathbb{P}^\star(\cdot|s,a))|\ln\left(\frac{|\mathcal{S}||\mathcal{A}|tH}{\delta}\right)}{N^t_{h(s)}(s,a)}$$

Inequality $(i)$ follows because $\|f\|_\infty \leq B$. Inequality $(ii)$ is a result of Cauchy-Schwarz, $(iii)$ uses the inequality $ab \leq \frac{a^2+b^2}{2}$ and $(iv)$ uses the condition $f \in [0,B]$ again. The above display implies,

$$\left|\left(\widehat{\mathbb{P}}_t(\cdot|s,a) - \mathbb{P}^\star(\cdot|s,a)\right)f\right| \leq \frac{3|\text{support}(\mathbb{P}^\star(\cdot|s,a))|BH\ln\left(\frac{|\mathcal{S}||\mathcal{A}|tH}{\delta}\right)}{N^t_{h(s)}(s,a)} + \frac{\sum_{s'\in\mathcal{S}}\mathbb{P}^\star(s'|s,a)f(s')}{H}.$$

Finally, since $f(s) \in [0,B]$,

$$\left|\left(\widehat{\mathbb{P}}_t(\cdot|s,a) - \mathbb{P}^\star(\cdot|s,a)\right)f\right| \leq B.$$

Combining these last two equations we conclude,

$$\left|\left(\widehat{\mathbb{P}}_t(\cdot|s,a) - \mathbb{P}^\star(\cdot|s,a)\right)f\right| \leq \min\left(B, \frac{3|\text{support}(\mathbb{P}^\star(\cdot|s,a))|BH\ln\left(\frac{|\mathcal{S}||\mathcal{A}|tH}{\delta}\right)}{N^t_{h(s)}(s,a)}\right) + \frac{\sum_{s'\in\mathcal{S}}\mathbb{P}^\star(s'|s,a)f(s')}{H}.$$

The result follows. □

From now on we'll use the notation

$$\xi_h^t(s,a) := (\beta - 1)\widetilde{V}^{\max} \min\left(\frac{3\,|\mathrm{support}(\mathbb{P}^\star(\cdot|s,a))|\,H\ln\left(\frac{|\mathcal{S}||\mathcal{A}|tH}{\delta}\right)}{N_h^t(s,a)}, 1\right).$$

Recall that by Assumption 3, if we define $h(s)$ as to horizon index of the state partitions that contains state $s$, we have $N_h(s,a) = 0$ for all $h \neq h(s)$.

The following corollary holds,

**Corollary B.7.** *With probability at least $1 - 2\delta$ for all $t \in \mathbb{N}, s \in \mathcal{S}, a \in \mathcal{A}, h \in [H]$ we have,*

$$\left(\widehat{\mathbb{P}}_t(\cdot|s,a) - \mathbb{P}^\star(\cdot|s,a)\right)\left(\widehat{V}_{h+1}^{\pi_t} - V_{h+1}^\star\right) \leq \frac{\mathbb{E}_{s' \sim \mathbb{P}^\star(\cdot|s,a)}\left[\widehat{V}_{h+1}^{\pi_t}(s') - V_{h+1}^\star(s')\right]}{H} + \xi_h^t(s,a)$$

*Proof.* Under the event of Lemma 5.1 (optimism), it follows that $\widehat{V}_{h+1}^{\pi_t}(s') \geq V_{h+1}^\star(s')$ for all $t \in \mathbb{N}, s \in \mathcal{S}$. Moreover, $\widehat{V}_{h+1}^{\pi_t}(s') - V_{h+1}^\star(s') \leq \beta\widetilde{V}_{h+1}(s') - V_{h+1}^\star(s') \leq (\beta - 1)\widetilde{V}_{h+1}(s')$.

Thus we can set $B = (\beta - 1)\widetilde{V}^{\max}$ in Lemma B.6. □

We will now restate a modified version of Lemma 7.10 from [2] that provides us a bound for term I in terms of expectations over sums of $b_h^t$ and $\xi_h^t$ terms.

**Lemma B.8** (Notation Adapted version of Lemma 7.10 from [2])**.** *For all $T \in \mathbb{N}$ with probability at least $1 - 3\delta$,*

$$\mathrm{I} \leq e\sum_{t=1}^T \mathbb{E}_{\tau \sim \pi_t}\left[\sum_{h=1}^H b_h^t(s_h,a_h) + \xi_h^t(s_h,a_h)\right].$$

Thus, with probability at least $1 - 3\delta$,

$$\sum_{t=1}^T V^\star(s_0) - V^{\pi_t}(s_0) \leq e\sum_{t=1}^T \mathbb{E}_{\tau \sim \pi_t}\left[\sum_{h=1}^H 2b_h^t(s_h,a_h) + \xi_h^t(s_h,a_h)\right].$$

The following Lemma will allow us to change from a sum of expectations over the played policies to a sum over the sample trajectories.

**Lemma B.9.** *Simultaneously for all $\mathcal{U} \subset \mathcal{S} \times \mathcal{A}$ and all $T \in \mathbb{N}$.*

$$\sum_{t=1}^T V^\star(s_0) - V^{\pi_t}(s_0) \leq e\sum_{t=1}^T \sum_{h=1}^H \mathbf{1}((s_h^t, a_h^t) \in \mathcal{U})\left(2b_h^t(s_h^t, a_h^t) + \xi_h^t(s_h^t, a_h^t)\right) +$$

$$e\sum_{t=1}^T \sum_{h=1}^H \mathbf{1}((s_h^t, a_h^t) \notin \mathcal{U})\left(2b_h^t(s_h^t, a_h^t) + \xi_h^t(s_h^t, a_h^t)\right) +$$

$$\mathcal{O}\left(\beta\widetilde{V}^{\max}\sqrt{HT\ln\left(\frac{T}{\delta}\right)}\right)$$

*and*

$$\sum_{t=1}^T V^\star(s_0) - V^{\pi_t}(s_0) \leq e\sum_{t=1}^T \mathbf{1}(\tau_t \cap \mathcal{U} = \emptyset)\sum_{h=1}^H \left(2b_h^t(s_h^t, a_h^t) + \xi_h^t(s_h^t, a_h^t)\right) +$$

$$\mathcal{O}\left(\beta\widetilde{V}^{\max}\sqrt{HT\ln\left(\frac{T}{\delta}\right)} + \beta H\widetilde{V}^{\max}\sum_{t=1}^T \mathbf{1}(\tau_t \cap \mathcal{U} \neq \emptyset)\right)$$

*With probability at least $1 - 4\delta$.*

*Proof.* By Lemma B.8, for all $T \in \mathbb{N}$ with probability at least $1 - 3\delta$,

$$\sum_{t=1}^{T} V^{\star}(s_0) - V^{\pi_t}(s_0) \leq e \sum_{t=1}^{T} \mathbb{E}_{\tau \sim \pi_t} \left[ \sum_{h=1}^{H} 2b_h^t(s_h, a_h) + \xi_h^t(s_h, a_h) \right]. \tag{10}$$

Since $|2b_h^t(s_h, a_h) + \xi_h^t(s_h, a_h)| \leq \mathcal{O}\left(\beta \widetilde{V}^{\max}\right)$, anytime Hoeffding lemma C.1 and the union bound implies that for all $T \in \mathbb{N}$ with probability at least $1 - 4\delta$,

$$\sum_{t=1}^{T} V^{\star}(s_0) - V^{\pi_t}(s_0) \leq e \sum_{t=1}^{T} \sum_{h=1}^{H} 2b_h^t(s_h^t, a_h^t) + \xi_h^t(s_h^t, a_h^t) + \mathcal{O}\left(\beta \widetilde{V}^{\max} \sqrt{HT \ln\left(\frac{T}{\delta}\right)}\right) \tag{11}$$

Observe that for any $\mathcal{U} \in \mathcal{S} \times \mathcal{A}$,

$$\begin{aligned}
\sum_{t=1}^{T} \sum_{h=1}^{H} 2b_h^t(s_h^t, a_h^t) + \xi_h^t(s_h^t, a_h^t) &= \sum_{t=1}^{T} \mathbf{1}(\tau_t \cap \mathcal{U} = \emptyset) \sum_{h=1}^{H} \left(2b_h^t(s_h^t, a_h^t) + \xi_h^t(s_h^t, a_h^t)\right) + \\
&\quad \sum_{t=1}^{T} \mathbf{1}(\tau_t \cap \mathcal{U} \neq \emptyset) \sum_{h=1}^{H} \left(2b_h^t(s_h^t, a_h^t) + \xi_h^t(s_h^t, a_h^t)\right) \\
&\leq \sum_{t=1}^{T} \mathbf{1}(\tau_t \cap \mathcal{U} = \emptyset) \sum_{h=1}^{H} \left(2b_h^t(s_h^t, a_h^t) + \xi_h^t(s_h^t, a_h^t)\right) + \\
&\quad \mathcal{O}\left(\beta H \widetilde{V}^{\max} \sum_{t=1}^{T} \mathbf{1}(\tau_t \cap \mathcal{U} \neq \emptyset)\right). \tag{12}
\end{aligned}$$

The result follows by combining inequalities 10, 11 and 12.

$\square$

## B.1 Full Proof of Theorem 5.2

We are ready to prove our main supporting Lemma,

**Lemma B.10.** *With probability at east $1 - 6\delta$ for all $\Delta > 0$ and $T \in \mathbb{N}$ simultaneously,*

$$\sum_{t=1}^{T} V^\star(s_0) - V^{\pi_t}(s_0) \le \beta H \widetilde{V}^{\max} \sqrt{|\mathcal{U}^{\text{good}}| T \ln \frac{|\mathcal{S}||\mathcal{A}|T}{\delta}} +$$

$$\mathcal{O}\Bigg( \min \Bigg( (\beta - 1) \widetilde{V}^{\max} |\mathcal{S}|^2 |\mathcal{A}| H^2 \ln \left( \frac{|\mathcal{S}||\mathcal{A}|TH}{\delta} \right) \log(T+1) +$$

$$\beta H \widetilde{V}^{\max} \sqrt{|\mathcal{U}^{\text{good}}| T \ln \frac{2|\mathcal{S}||\mathcal{A}|T}{\delta}} +$$

$$\beta \left( \widetilde{V}^{\max} \right)^2 \left( \ln \frac{|\mathcal{S}||\mathcal{A}|T}{\delta} \right) \frac{\sqrt{H|\mathcal{S}||\mathcal{A}||\text{BoundaryPseudoSub}_\Delta|}}{\Delta} +$$

$$\beta \widetilde{V}^{\max} |\mathcal{S}||\mathcal{A}| \ln \frac{|\mathcal{S}||\mathcal{A}|T}{\delta} \ln(T+1),$$

$$(\beta - 1) \widetilde{V}^{\max} |\mathcal{S} \backslash \text{PathPseudoSub}_\Delta| |\mathcal{U}^{\text{good}}| H^2 \ln \left( \frac{|\mathcal{S}||\mathcal{A}|TH}{\delta} \right) \log(T+1) +$$

$$\beta \widetilde{V}^{\max} |\mathcal{U}^{\text{good}}| \ln \frac{|\mathcal{S}||\mathcal{A}|T}{\delta} \ln(T+1) +$$

$$\beta H \widetilde{V}^{\max} \sqrt{|\mathcal{U}^{\text{good}}| T \ln \frac{|\mathcal{S}||\mathcal{A}|T}{\delta}} +$$

$$\beta^3 H \left( \widetilde{V}^{\max} \right)^3 |\text{BoundaryPseudoSub}_\Delta| \times \frac{\ln \frac{|\mathcal{S}||\mathcal{A}|T}{\delta}}{\Delta^2} \Bigg) \Bigg)$$

*Where* $\mathcal{U}^{\text{good}} = (\mathcal{S} \times \mathcal{A}) \backslash (\text{PathPseudoSub}_\Delta \times \mathcal{A} \cup \text{PseudoSub}_\Delta).$

*Proof.* We will condition on the events from Lemmas B.4 and B.9, something that happens with probability at least $1 - 6\delta$.

We'll start with the decomposition from Lemma B.9. Set $\mathcal{U}^{\text{good}} = (\mathcal{S} \times \mathcal{A}) \backslash [(\text{PathPseudoSub}_\Delta \times \mathcal{A}) \cup \text{PseudoSub}_\Delta]$. In what follows we'll use $\mathcal{U}^{\text{good}}$ to denote this set when convenient. We will also use the notation $\mathcal{U}^{\text{bad}} = (\mathcal{S} \times \mathcal{A}) \backslash \mathcal{U}^{\text{good}}$ to denote the complement of $\mathcal{U}_{\text{good}}$.

Recall that as a consequence of Equation 4 for all state action pairs $(s,a) \in \mathcal{U}^{\text{good}}$, we have that $\text{Neighbor}(s,a) \subseteq \mathcal{S} \backslash \text{PathPseudoSub}_\Delta$.

Therefore, the support of $\mathbb{P}^\star(\cdot|s,a)$ and $\widehat{\mathbb{P}}_t(\cdot|s,a)$ is contained in $\mathcal{S} \backslash \text{PathPseudoSub}_\Delta$ for $(s,a) \in \mathcal{S} \backslash (\text{PathPseudoSub}_\Delta \times \mathcal{A} \cup \text{PseudoSub}_\Delta)$. This implies that for all $(s,a) \in (\mathcal{S} \times \mathcal{A}) \backslash (\text{PathPseudoSub}_\Delta \times \mathcal{A} \cup \text{PseudoSub}_\Delta)$, we have

$$\xi_h^t(s,a) \le (\beta - 1) \widetilde{V}^{\max} \min \left( \frac{3 |\mathcal{S} \backslash \text{PathPseudoSub}_\Delta| H \ln \left( \frac{|\mathcal{S}||\mathcal{A}|tH}{\delta} \right)}{N_h^t(s,a)}, 1 \right) \quad \forall \, (s,a) \in \mathcal{U}^{\text{good}}. \quad (13)$$

Similarly,

$$\xi_h^t(s,a) \le (\beta - 1) \widetilde{V}^{\max} \min \left( \frac{3 |\mathcal{S}| H \ln \left( \frac{|\mathcal{S}||\mathcal{A}|tH}{\delta} \right)}{N_h^t(s,a)}, 1 \right) \quad \forall \, (s,a) \in \mathcal{U}^{\text{bad}}. \quad (14)$$

Let $\mathcal{U}^{\text{good}} = (\mathcal{S} \times \mathcal{A}) \backslash (\text{PathPseudoSub}_\Delta \times \mathcal{A} \cup \text{PseudoSub}_\Delta)$. Inequality 13 implies,

$$\sum_{t=1}^{T} \sum_{h=1}^{H} \mathbf{1}((s_h^t, a_h^t) \in \mathcal{U}^{\text{good}}) \xi_h^t(s_h^t, a_h^t) \tag{15}$$

$$\leq \sum_{t=1}^{T} \sum_{h=1}^{H} \mathbf{1}((s_h^t, a_h^t) \in \mathcal{U}^{\text{good}})(\beta - 1)\widetilde{V}^{\max} \min\left( \frac{3|\mathcal{S} \backslash \text{PathPseudoSub}_\Delta| H \ln\left(\frac{|\mathcal{S}||\mathcal{A}|tH}{\delta}\right)}{N_h^t(s_h^t, a_h^t)}, 1 \right)$$

$$\leq 3(\beta - 1)\widetilde{V}^{\max} |\mathcal{S} \backslash \text{PathPseudoSub}_\Delta| H \ln\left(\frac{|\mathcal{S}||\mathcal{A}|TH}{\delta}\right) \sum_{t=1}^{T} \sum_{h=1}^{H} \frac{\mathbf{1}((s_h^t, a_h^t) \in \mathcal{U}^{\text{good}})}{N_h^t(s_h^t, a_h^t)}$$

$$\overset{(i)}{\leq} 6(\beta - 1)\widetilde{V}^{\max} |\mathcal{S} \backslash \text{PathPseudoSub}_\Delta| |\mathcal{U}^{\text{good}}| H^2 \ln\left(\frac{|\mathcal{S}||\mathcal{A}|TH}{\delta}\right) \log(T+1).$$

Where inequality $(i)$ holds because of Lemma C.5. The exact same argument applied to the upper bound of Equation 15 implies,

$$\sum_{t=1}^{T} \sum_{h=1}^{H} \mathbf{1}((s_h^t, a_h^t) \in \mathcal{U}^{\text{bad}}) \xi_h^t(s_h^t, a_h^t) \leq 6(\beta - 1)\widetilde{V}^{\max} |\mathcal{S}| |\mathcal{U}^{\text{bad}}| H^2 \ln\left(\frac{|\mathcal{S}||\mathcal{A}|TH}{\delta}\right) \log(T+1)$$

$$\leq 6(\beta - 1)\widetilde{V}^{\max} |\mathcal{S}|^2 |\mathcal{A}| H^2 \ln\left(\frac{|\mathcal{S}||\mathcal{A}|TH}{\delta}\right) \log(T+1). \tag{16}$$

Let's now bound the sum $\sum_{t=1}^{T} \sum_{h=1}^{H} \mathbf{1}((s_h^t, a_h^t) \in \mathcal{U}^{\text{good}}) b_h^t(s_h^t, a_h^t)$.

$$\sum_{t=1}^{T} \sum_{h=1}^{H} \mathbf{1}((s_h^t, a_h^t) \in \mathcal{U}^{\text{good}}) b_h^t(s_h^t, a_h^t) \tag{17}$$

$$= \sum_{t=1}^{T} \sum_{h=1}^{H} \mathbf{1}((s_h^t, a_h^t) \in \mathcal{U}^{\text{good}}) \min\left( 16\beta \sqrt{\frac{\widehat{\mathbb{E}}_{s' \sim \widehat{\mathbb{P}}_t(\cdot | s_h^t, a_h^t)}[\widetilde{V}_{h+1}^2(s') | s_h^t, a_h^t] \ln\frac{2|\mathcal{S}||\mathcal{A}|}{\delta}}{N_h^t(s_h^t, a_h^t)}} + \frac{12\beta \widetilde{V}^{\max}}{N_h^t(s_h^t, a_h^t)} \ln\frac{2|\mathcal{S}||\mathcal{A}|t}{\delta}, 2\beta \widetilde{V}^{\max} \right)$$

$$\leq \sum_{t=1}^{T} \sum_{h=1}^{H} \mathbf{1}((s_h^t, a_h^t) \in \mathcal{U}^{\text{good}}) \left( 16\beta \sqrt{\frac{\widehat{\mathbb{E}}_{s' \sim \widehat{\mathbb{P}}_t(\cdot | s_h^t, a_h^t)}[\widetilde{V}_{h+1}^2(s') | s_h^t, a_h^t] \ln\frac{2|\mathcal{S}||\mathcal{A}|}{\delta}}{N_h^t(s_h^t, a_h^t)}} + \frac{12\beta \widetilde{V}^{\max}}{N_h^t(s_h^t, a_h^t)} \ln\frac{2|\mathcal{S}||\mathcal{A}|t}{\delta} \right)$$

$$\overset{(a)}{\leq} 16\beta \widetilde{V}^{\max} \sqrt{\ln\frac{2|\mathcal{S}||\mathcal{A}|T}{\delta}} \sum_{t=1}^{T} \sum_{h=1}^{H} \frac{\mathbf{1}((s_h^t, a_h^t) \in \mathcal{U}^{\text{good}})}{\sqrt{N_h^t(s_h^t, a_h^t)}} + 12\beta \widetilde{V}^{\max} \ln\frac{2|\mathcal{S}||\mathcal{A}|T}{\delta} \sum_{t=1}^{T} \sum_{h=1}^{H} \frac{\mathbf{1}((s_h^t, a_h^t) \in \mathcal{U}^{\text{good}})}{N_h^t(s_h^t, a_h^t)}$$

$$\overset{(i)}{\leq} 32\beta H \widetilde{V}^{\max} \sqrt{|\mathcal{U}^{\text{good}}| T \ln\frac{2|\mathcal{S}||\mathcal{A}|T}{\delta}} + 24\beta \widetilde{V}^{\max} |\mathcal{U}^{\text{good}}| \ln\frac{2|\mathcal{S}||\mathcal{A}|T}{\delta} \ln(T+1).$$

Where inequality $(i)$ holds because of Lemmas C.4 and C.5. The exact same proof argument as in the proof of inequality 17 that leads to inequality $(a)$ yields,

$$\sum_{t=1}^{T}\sum_{h=1}^{H}\mathbf{1}((s_h^t,a_h^t)\in\mathcal{U}^{\text{bad}})b_h^t(s_h^t,a_h^t) \tag{18}$$

$$\leq 16\beta\widetilde{V}^{\max}\sqrt{\ln\frac{2|\mathcal{S}||\mathcal{A}|T}{\delta}\sum_{t=1}^{T}\sum_{h=1}^{H}\frac{\mathbf{1}((s_h^t,a_h^t)\in\mathcal{U}^{\text{bad}})}{\sqrt{N_h^t(s_h^t,a_h^t)}}}+12\beta\widetilde{V}^{\max}\ln\frac{2|\mathcal{S}||\mathcal{A}|T}{\delta}\sum_{t=1}^{T}\sum_{h=1}^{H}\frac{\mathbf{1}((s_h^t,a_h^t)\in\mathcal{U}^{\text{bad}})}{N_h^t(s_h^t,a_h^t)}$$

$$\overset{(i)}{\leq} 32\beta\widetilde{V}^{\max}\sqrt{H|\mathcal{U}^{\text{bad}}|\ln\frac{2|\mathcal{S}||\mathcal{A}|T}{\delta}\sum_{(s,a)\in(\mathcal{S}\times\mathcal{A})\backslash\mathcal{U}^{\text{good}}}\sum_{h=1}^{H}N_h^T(s,a)}+24\beta\widetilde{V}^{\max}|\mathcal{U}^{\text{bad}}|\ln\frac{2|\mathcal{S}||\mathcal{A}|T}{\delta}\ln(T+1)$$

$$\overset{(ii)}{\leq} 32\times 8192\beta\left(\widetilde{V}^{\max}\right)^2\left(\ln\frac{2|\mathcal{S}||\mathcal{A}|T}{\delta}\right)\frac{\sqrt{H|\mathcal{U}^{\text{bad}}||\text{BoundaryPseudoSub}_\Delta|}}{\Delta}+$$

$$24\beta\widetilde{V}^{\max}|\mathcal{U}^{\text{bad}}|\ln\frac{2|\mathcal{S}||\mathcal{A}|T}{\delta}\ln(T+1)$$

$$=\mathcal{O}\left(\beta\left(\widetilde{V}^{\max}\right)^2\left(\ln\frac{2|\mathcal{S}||\mathcal{A}|T}{\delta}\right)\frac{\sqrt{H|\mathcal{S}||\mathcal{A}||\text{BoundaryPseudoSub}_\Delta|}}{\Delta}\right)+$$

$$\mathcal{O}\left(\beta\widetilde{V}^{\max}|\mathcal{S}||\mathcal{A}|\ln\frac{2|\mathcal{S}||\mathcal{A}|T}{\delta}\ln(T+1)\right),$$

where inequality (*i*) follows because of Lemmas C.4 and C.5. And inequality (*ii*) follows from Lemma B.4.

Combining inequalities 15, 16, 17 and 18 with Lemma B.9 we get,

$$\sum_{t=1}^{T}V^\star(s_0)-V^{\pi_t}(s_0)\leq e\sum_{t=1}^{T}\sum_{h=1}^{H}\mathbf{1}((s_h^t,a_h^t)\in\mathcal{U}^{\text{good}})\left(2b_h^t(s_h^t,a_h^t)+\xi_h^t(s_h^t,a_h^t)\right)+$$

$$e\sum_{t=1}^{T}\sum_{h=1}^{H}\mathbf{1}((s_h^t,a_h^t)\in\mathcal{U}^{\text{bad}})\left(2b_h^t(s_h^t,a_h^t)+\xi_h^t(s_h^t,a_h^t)\right)+\mathcal{O}\left(\beta\widetilde{V}^{\max}\sqrt{HT\ln\left(\frac{T}{\delta}\right)}\right)$$

$$\leq\mathcal{O}\left((\beta-1)\widetilde{V}^{\max}|\mathcal{S}|^2|\mathcal{A}|H^2\ln\left(\frac{|\mathcal{S}||\mathcal{A}|TH}{\delta}\right)\log(T+1)\right)+$$

$$\mathcal{O}\left(\beta H\widetilde{V}^{\max}\sqrt{|\mathcal{U}^{\text{good}}|T\ln\frac{2|\mathcal{S}||\mathcal{A}|T}{\delta}}\right)+$$

$$\mathcal{O}\left(\beta\left(\widetilde{V}^{\max}\right)^2\left(\ln\frac{|\mathcal{S}||\mathcal{A}|T}{\delta}\right)\frac{\sqrt{H|\mathcal{S}||\mathcal{A}||\text{BoundaryPseudoSub}_\Delta|}}{\Delta}\right)+$$

$$\mathcal{O}\left(\beta\widetilde{V}^{\max}|\mathcal{S}||\mathcal{A}|\ln\frac{|\mathcal{S}||\mathcal{A}|T}{\delta}\ln(T+1)\right)$$

We will now derive a bound that has only logarithmic dependence on $\mathcal{S}$ at the cost of a quadratic dependence on $\Delta^2$.

By Lemma B.9, inequalities 15 and 17 and Lemma B.4.

$$\sum_{t=1}^{T} V^{\star}(s_0) - V^{\pi_t}(s_0) \le e \sum_{t=1}^{T} \mathbf{1}(\tau_t \cap \mathcal{U}^{\text{bad}} = \emptyset) \sum_{h=1}^{H} \left(2b_h^t(s_h^t, a_h^t) + \xi_h^t(s_h^t, a_h^t)\right) +$$

$$\mathcal{O}\left(\beta \widetilde{V}^{\max} \sqrt{HT \ln\left(\frac{T}{\delta}\right)} + \beta H \widetilde{V}^{\max} \sum_{t=1}^{T} \mathbf{1}(\tau_t \cap \mathcal{U} \ne \emptyset)\right)$$

$$\le e \sum_{t=1}^{T} \sum_{h=1}^{H} \mathbf{1}((s_h^t, a_h^t) \in \mathcal{U}^{\text{good}}) \left(2b_h^t(s_h^t, a_h^t) + \xi_h^t(s_h^t, a_h^t)\right) +$$

$$\mathcal{O}\left(\beta \widetilde{V}^{\max} \sqrt{HT \ln\left(\frac{T}{\delta}\right)} + \beta H \widetilde{V}^{\max} \sum_{t=1}^{T} \mathbf{1}(\tau_t \cap \mathcal{U} \ne \emptyset)\right)$$

$$\overset{(i)}{=} \mathcal{O}\left((\beta - 1)\widetilde{V}^{\max} |\mathcal{S} \backslash \text{PathPseudoSub}_\Delta| |\mathcal{U}^{\text{good}}| H^2 \ln\left(\frac{|\mathcal{S}||\mathcal{A}|TH}{\delta}\right) \log(T+1)\right) +$$

$$\mathcal{O}\left(\beta \widetilde{V}^{\max} |\mathcal{U}^{\text{good}}| \ln \frac{|\mathcal{S}||\mathcal{A}|T}{\delta} \ln(T+1)\right) +$$

$$\mathcal{O}\left(\beta H \widetilde{V}^{\max} \sqrt{|\mathcal{U}^{\text{good}}| T \ln \frac{|\mathcal{S}||\mathcal{A}|T}{\delta}}\right) +$$

$$\mathcal{O}\left(\beta^3 H \left(\widetilde{V}^{\max}\right)^3 |\text{BoundaryPseudoSub}_\Delta| \times \frac{\ln \frac{|\mathcal{S}||\mathcal{A}|T}{\delta}}{\Delta^2}\right)$$

This holds for all $\Delta \in [0,1]$ simultaneously because Lemma B.4 holds simultaneously for all $\Delta \in [0,1]$ and for all $T \in \mathbb{N}$ because all the events we have conditioned refer to properties that hold for all $T \in \mathbb{N}$. This finalizes the proof of the desired result. $\qquad\square$

Using the bound $|\mathcal{U}^{\text{good}}| \le |\mathcal{S} \backslash \text{PathPseudoSub}_\Delta| |\mathcal{A}|$, we have the following version of our results

**Theorem B.11** (Full version of Theorem 5.2). *The regret of UCBVI - Shaped satisfies,*

$$\sum_{t=1}^{T} V^{\star}(s_0) - V^{\pi_t}(s_0) = \mathcal{O}\left(\min_\Delta \left(H\beta \widetilde{V}^{\max} \sqrt{|\mathcal{S} \backslash \text{PathPseudoSub}_\Delta| |\mathcal{A}| T \ln \frac{\widetilde{V}^{\max} |\mathcal{S}||\mathcal{A}|T}{\delta}} + \right.\right.$$

$$\left.\left. \beta \widetilde{V}^{\max} \ln \frac{\widetilde{V}^{\max} |\mathcal{S}||\mathcal{A}|T}{\delta} \times \min\left(\bar{A}(\Delta), \bar{B}(\Delta)\right)\right)\right).$$

*For all $T \in \mathbb{N}$ with probability at least $1 - 6\delta$. Where*

$$\bar{A}(\Delta) = \frac{\beta \widetilde{V}^{\max} H^{1/2} |\mathcal{S}|^{1/2} |\mathcal{A}|^{1/2} |\text{BoundaryPseudoSub}_\Delta|^{1/2}}{\Delta} + \frac{(\beta-1)}{\beta} |\mathcal{S}|^2 |\mathcal{A}| H^2 \log(T+1)$$

$$+ |\mathcal{S}||\mathcal{A}| \ln(T+1),$$

*and*

$$\bar{B}(\Delta) = \frac{\beta^2 \left(\widetilde{V}^{\max}\right)^2 H |\text{BoundaryPseudoSub}_\Delta|}{\Delta^2} + \frac{(\beta-1)}{\beta} |\mathcal{S} \backslash \text{PathPseudoSub}_\Delta|^2 |\mathcal{A}| H^2 \log(T+1)$$

$$+ |\mathcal{S} \backslash \text{PathPseudoSub}_\Delta| |\mathcal{A}| \ln(T+1).$$

## B.2 Generalizations

In this section we briefly discuss what can we say in the case the shaping functions $\{\widetilde{Q}_h\}_{h \in [H]}$ are of the form $\widetilde{Q}_h : \mathcal{S} \times \mathcal{A} \to \mathbb{R}$ such that $Q_h^{\star}(s,a) \le \beta \widetilde{Q}_h(s,a)$ instead of our state-only assumption. By doing so we recover some of the results of [21] for which we provide simple proofs. We will use the notation $\widetilde{Q}^{\max} = \max_{s,a,h} \widetilde{Q}_h(s,a)$.

Let's consider a generic optimistic RL algorithm $\mathbb{A}$ that computes an *optimistic Q* function estimator $\{\widehat{Q}_h^t\}_{h\in[H]}$ at the beginning of time-step $t$ satisfying $\widehat{Q}_h^t(s,\pi_\star(s)) \geq Q_\star(s,\pi_\star(s))$. Notice that we only consider optimism at the optimal action $\pi_\star(s)$. Let's also assume the policy played by $\mathbb{A}$ at time $t$ equals $\pi_t(\cdot|s,a,h) = \arg\max_{a\in\mathcal{A}} \widehat{Q}_h^t(s,a)$. We will introduce the notion of a least upper bound stability for optimistic algorithms.

**Definition 1.** *We say that an optimistic algorithm $\mathbb{A}$ is stable w.r.t. to clipping if substituting the $\{\widetilde{Q}_h^t\}_{h\in[H]}$ values by their clipped versions*

$$\widehat{Q}_h^{t,\mathrm{clipped}}(s,a) \leftarrow \min\left(\widehat{Q}_h^t(s,a), \beta\widetilde{Q}(s,a)\right)$$

*does not affect its regret guarantees.*

Clipping is done *after* the $\widehat{Q}_h^t(s,a)$ values have been computed.

**Optimism is preserved.** When the un-clipped $\{\widehat{Q}_h^t\}_{h\in[H]}$ are optimistic, then as long as $\beta\widetilde{Q}_h(s,\pi_\star(s)) \geq Q_\star(s,\pi_\star(s))$, the maximum among all the clipped values is also optimistic.

$$\max_{a\in\mathcal{A}} \widehat{Q}_h^{t,\mathrm{clipped}}(s,a) = \max_{a\in\mathcal{A}}\left[\min\left(\widehat{Q}_h^t(s,a), \beta\widetilde{Q}(s,a)\right)\right] \geq \min\left(\widehat{Q}_h^t(s,\pi_\star(s)), \beta\widetilde{Q}(s,\pi_\star(s))\right) \geq Q^\star(s,\pi_\star(s))$$

If $\mathbb{A}$'s policy at time $t$ equals $\pi_t(\cdot|s,h) = \arg\max_{a\in\mathcal{A}} \widehat{Q}_h^{t,\mathrm{clipped}}(s,a)$, as long as the un-clipped $\{\widehat{Q}_h^t\}_{h\in[H]}$ are optimistic the support of policy $\pi_t$ satisfies

$$\mathrm{Support}(\pi_t(\cdot|s,a,h)) = \{a' \text{ s.t. } \beta\widetilde{Q}_h(s,a') \geq Q_h^\star(s,\pi_\star(s,h))\}$$

This is simply because for all $a \in \mathcal{A}\backslash\{a' \text{ s.t. } \beta\widetilde{Q}_h(s,a') \geq Q_h(s,\pi_\star(s,h)\}$, the clipped $\{\widehat{Q}_h^{t,\mathrm{clipped}}(s,a)\}_{h\in[H]}$ satisfy

$$\widehat{Q}_h^{t,\mathrm{clipped}}(s,a) < Q_h^\star(s,\pi_\star(s,h))$$

In other words, for all $t \in \mathbb{N}$ the clipped algorithm will only visit state-action pairs in $\cup_{s\in\mathcal{S},h\in[H]}\{a \text{ s.t. } \beta\widetilde{Q}_h(s,a) \geq Q_h^\star(s,\pi_\star(s,h))\}$. Alternatively this can be thought of as if the learner were to interact only with a prunned MDP with state space $\mathcal{S}$ and state (and horizon) dependent action sets $\mathcal{A}(s)$ of the form $\mathcal{A}(s) = \{a \text{ s.t. } \beta\widetilde{Q}_{h(s)}(s,a) \geq Q_{h(s)}^\star(s,\pi_\star(s,h(s)))\}$.

It is easy to see the corresponding version of UCBVI-Shaped satisfies the following regret guarantee that recovers the min-max rates of Theorem 1.1 in [21],

**Lemma B.12.** *The regret of UCBVI-Shaped with $\widetilde{Q}$-shaping satisfies,*

$$\sum_{t=1}^T V^\star(s_0) - V^{\pi_t}(s_0) \leq \mathcal{O}\left(\beta\widetilde{Q}^{\max}\sqrt{|\mathrm{Pair}_{\mathrm{eff}}|T\ln\frac{|\mathcal{S}||\mathcal{A}|T}{\delta}}\right) +$$

$$\mathcal{O}\left(\beta\widetilde{Q}^{\max}H|\mathcal{S}||\mathrm{Pair}_{\mathrm{eff}}|\ln\frac{|\mathcal{S}||\mathcal{A}|T}{\delta}\ln(T+1)\right)$$

*For all $T \in \mathbb{N}$ with probability at least $1 - \delta$, where[8] $\mathrm{Pair}_{\mathrm{eff}} = \{(s,a) \in \mathcal{S}\times\mathcal{A} \text{ s.t. } a \in \mathcal{A}(s)\}$.*

*Proof.* We borrow the bound from Theorem 7.6 in [2]. Observe that as long as optimism holds (Lemma 5.1), the regret decomposition of Lemma B.9 is satisfied.

$$\sum_{t=1}^T V^\star(s_0) - V^{\pi_t}(s_0) \leq e\sum_{t=1}^T\sum_{h=1}^H \left(2b_h^t(s_h^t,a_h^t) + \xi_h^t(s_h^t,a_h^t)\right) + \mathcal{O}\left(\beta\widetilde{Q}^{\max}\sqrt{HT\ln\left(\frac{T}{\delta}\right)}\right)$$

Notice that as long as optimism holds, for all $s_h^t$ the action $a_h^t$ satisfies $a_h^t \in \mathcal{A}(s_h^t)$. Thus,

---

[8]Recall we have assumed that $\mathcal{S}$ is $h$ indexed so that any state $s \in \mathcal{S}$ can be accessed only during in-episode time-step $h(s)$.

$$\sum_{t=1}^{T}\sum_{h=1}^{H}\xi_h^t(s_h^t,a_h^t) \leq \sum_{t=1}^{T}\sum_{h=1}^{H}\beta\widetilde{Q}^{\max}\min\left(\frac{3\,|\mathcal{S}|\,H\ln\left(\frac{|\mathcal{S}||\mathcal{A}|tH}{\delta}\right)}{N_h^t(s_h^t,a_h^t)},1\right)$$

$$\leq 3\beta\widetilde{Q}^{\max}\,|\mathcal{S}|\,H\ln\left(\frac{|\mathcal{S}||\mathcal{A}|TH}{\delta}\right)\sum_{t=1}^{T}\sum_{h=1}^{H}\frac{1}{N_h^t(s_h^t,a_h^t)}$$

$$\overset{(i)}{\leq} 6\beta\widetilde{Q}^{\max}\,|\mathcal{S}|\,\mathrm{Pair}_{\mathrm{eff}}\cdot H\ln\left(\frac{|\mathcal{S}||\mathcal{A}|TH}{\delta}\right)\log(T+1).$$

Where inequality $(i)$ follows from Lemma C.5. Similarly,

$$\sum_{t=1}^{T}\sum_{h=1}^{H}b_h^t(s_h^t,a_h^t) \leq 16\beta\widetilde{Q}^{\max}\sqrt{\ln\frac{2|\mathcal{S}||\mathcal{A}|T}{\delta}}\sum_{t=1}^{T}\sum_{h=1}^{H}\frac{1}{\sqrt{N_h^t(s_h^t,a_h^t)}}$$

$$+ 12\beta\widetilde{Q}^{\max}\ln\frac{2|\mathcal{S}||\mathcal{A}|T}{\delta}\sum_{t=1}^{T}\sum_{h=1}^{H}\frac{1}{N_h^t(s_h^t,a_h^t)}$$

$$\overset{(i)}{\leq} 32\beta\widetilde{Q}^{\max}\sqrt{|\mathrm{Pair}_{\mathrm{eff}}|T\ln\frac{2|\mathcal{S}||\mathcal{A}|T}{\delta}}$$

$$+ 24\beta\widetilde{Q}^{\max}|\mathrm{Pair}_{\mathrm{eff}}|\ln\frac{2|\mathcal{S}||\mathcal{A}|T}{\delta}\ln(T+1)$$

Where $(i)$ holds by Lemmas C.4 and C.5. Combining these last two inequalities yields

$$\sum_{t=1}^{T}V^\star(s_0)-V^{\pi_t}(s_0) \leq \mathcal{O}\left(\beta\widetilde{Q}^{\max}\sqrt{|\mathrm{Pair}_{\mathrm{eff}}|T\ln\frac{|\mathcal{S}||\mathcal{A}|T}{\delta}}\right) +$$

$$\mathcal{O}\left(\beta\widetilde{Q}^{\max}H\,|\mathcal{S}|\,|\mathrm{Pair}_{\mathrm{eff}}|\ln\frac{|\mathcal{S}||\mathcal{A}|T}{\delta}\ln(T+1)\right)$$

The result follows.

$\square$

Similarly, we can obtain a result for instance dependent bounds for a clipped version of the StrongEuler algorithm. We consider the exact same clipping mechanism as in the previous discussion. Optimism guarantees (just as we have described above) that $\mathrm{Support}(\pi_t(\cdot|s,a,h)) = \{a' \text{ s.t. } \beta\widetilde{Q}_h(s,a') \geq Q_h^\star(s,\pi_\star(s,h))\}$. Therefore the data generated by the clipped StrongEuler algorithm will be produced from the 'prunned' MDP with state space $\mathcal{S}$ and state-horizon dependent action sets $\mathcal{A}(s)$ (i.e. with state-action space $\mathrm{Pair}_{\mathrm{eff}}$). Strong optimism holds for the estimates $\widehat{Q}_h^t(s,a)$ for all $(s,a) \in \mathrm{Pair}_{\mathrm{eff}}$ and all time steps $t$ (notice that $\pi_\star$ is an optimal policy both in the original as well as in the pruned MDP). The exact same proofs as in [16] (see proof of Theorem 3.2) hold in this case with the only modification being to adapt the argument for MDPs where the number of actions is state-dependent thus yielding the following result,

**Lemma B.13.** *The regret of clipped-StrongEuler satisfies with high probability,*

$$\sum_{t=1}^{T}V^\star(s_0)-V^{\pi_t}(s_0) = \mathcal{O}\left(\sum_{(s,a)\in\mathrm{Pair}_{\mathrm{eff}}}\frac{Q_{h(s)}^\star(s,a)}{\mathrm{gap}_{h(s)}(s,a)}\log(T)\right)$$

*where* $\mathrm{gap}_h(s,a) = V_h^\star(s) - Q_h^\star(s,a)$

Stating this result in terms of the return gaps (see Definition 3.1 in [16]) over $\mathrm{Pair}_{\mathrm{eff}}$ is possible. Lemma B.13 recovers the instance dependent results of [21].

## B.3 Connections with instance dependent rates for reinforcement learning

Although our main results are not instance dependent, our rates can be much better if the gaps over the effective state space are very small and the effective state space is also very small. Moreover, in contrast with instance dependent rates our results depend on an effective state space size that may be much smaller than the original state space. Our main innovation lies in these state pruning results.

## B.4 Model Selection

Although UCBVI-Shaped requires knowledge of $\beta$, Algorithm 1 can be used to design a parameter free version. As explained in Section 6 we simply require to initialize algorithm 1 with a set of $N$ different $\beta$ parameter guesses $[\beta_1, \cdots, \beta_N]$. A good choice for these is an exponential parameter grid of the form $\beta_1 = 1, \cdots, \beta_N = 2^N$.

In the case we are using the Corral algorithm of [34] (and [1]), it is enough to set the learning rates based on a putative regret bound value of $\sqrt{T}$ (see the proof of Theorem 5.3 in [34] where the learning rate can be set to $\eta = \sqrt{\frac{N}{T}}$). The main term in the regret guarantee of Theorem 5.2 scales with $\sqrt{T}$. Thus, for any choice of $\beta_i$ if this value of $\beta_i$ was valid, the regret rate would scale as $C_i\sqrt{T\log(1/\delta)}$ for some parameter $C_i \in [1, H\beta_i\widetilde{V}^{\max}\sqrt{|\mathcal{S}||\mathcal{A}|}]$. If using regret Balancing as an online model selection mechanism (see [33]) we can use an exponential parameter grid of the form $\left\{\beta_i \times [1, 2, \cdots, H\beta_i\widetilde{V}^{\max}\sqrt{|\mathcal{S}||\mathcal{A}|}]\right\}_{i=1}^{N}$. The number of base algorithms necessary in this case is at most $N\log(H\beta_N\widetilde{V}^{\max}\sqrt{|\mathcal{S}||\mathcal{A}|})$.

As a consequence of Theorem 5.3 and as a simple corollary of Theorem 5.3 in [34] or Theorem 5.1 in [33] we conclude that,

**Lemma B.14.** *Provided that $\beta_N\widetilde{V}_h(s) \geq V_h^\star(s)$ for all $s \in \mathcal{S}$ and $h \in [H]$ (although it could be that $\beta_i\widetilde{V}_h(s) \geq V_h^\star(s)$ for all $s \in \mathcal{S}$ and $h \in [H]$ for $i \ll N$ ), the expected regret of Algorithm 1 satisfies,*

$$\mathbb{E}\left[\sum_{t=1}^{T} V^\star(s_0) - V^{\pi_t}(s_0)\right] = \widetilde{\mathcal{O}}\left(C^2\sqrt{NT}\right)$$

*when using Stochastic CORRAL as a model selection strategy. Similarly the expected regret of Algorithm 1 satisfies,*

$$\sum_{t=1}^{T} V^\star(s_0) - V^{\pi_t}(s_0) = \widetilde{\mathcal{O}}\left(C^2\sqrt{N\log(H\beta_N\widetilde{V}^{\max}\sqrt{|\mathcal{S}||\mathcal{A}|})T\log(1/\delta)}\right)$$

*with probability at least $1 - \mathcal{O}(\delta)$ when using Regret Balancing as a model selection strategy.*

*Proof.* This result is (almost) an immediate corollary of Theorems 5.3 and 5.1 in [34] and [34] respectively. In order to apply these results to our setting we only need to note that for the optimal value of $\beta$ (among the choices in $[\beta_1, \cdots, \beta_N]$) there exists a constant $C \in [1, 2, \cdots, H\beta_N\widetilde{V}^{\max}\sqrt{|\mathcal{S}||\mathcal{A}|}]$ such that regret rate of UCBVI-Shaped (Theorem 5.2) can be upper bounded as,

$$\sum_{t=1}^{T} V^\star(s_0) - V^{\pi_t}(s_0) = \widetilde{\mathcal{O}}\left(C\sqrt{T\log(1/\delta)}\right).$$

with probability at least $1 - \mathcal{O}(\delta)$ and where $\widetilde{\mathcal{O}}$ may hide polynomial factors in $|\mathcal{S}|$ and $|\mathcal{A}|$.

Applying the in-expectation results of Theorem 5.3 in [34] for the Stochastic CORRAL model selection algorithm yields (setting $\delta \ll \frac{1}{T}$),

$$\mathbb{E}\left[\sum_{t=1}^{T} V^\star(s_0) - V^{\pi_t}(s_0)\right] = \widetilde{\mathcal{O}}\left(C^2\sqrt{NT\log(1/\delta)}\right)$$

Where $\widetilde{\mathcal{O}}$ may hide polynomial factors in $|\mathcal{S}|$ and $|\mathcal{A}|$ and $T$. Similarly, Regret Balancing yields a bound,

$$\sum_{t=1}^{T} V^{\star}(s_0) - V^{\pi_t}(s_0) = \widetilde{\mathcal{O}}\left( C^2 \sqrt{N \log(H\beta_N \widetilde{V}^{\max} \sqrt{|\mathcal{S}||\mathcal{A}|})T \log(1/\delta)} \right)$$

with probability at least $1 - \mathcal{O}(\delta)$.

$\square$

The pseudocode for this algoritihm is provided below:

---

**Algorithm 3** Online UCBVI - Shaped

---

1: **Input** reward function $r$ (assumed to be known), set of $\beta$ values - $[\beta_1, \beta_2, \dots, \beta_N]$
2: Initialize model selection probability $p(\beta)$ as a uniform distribution over $[\beta_1, \beta_2, \dots, \beta_N]$
3: **for** $t = 1, \dots, T$
4:     Sample a value of $\beta \sim p(\beta)$
5:     Run UCBVI-Shaped ($\beta$) with this sampled beta
6:     Update $p(\beta)$ using samples from UCBVI-Shaped using an online model selection algorithm.
7: **End for**

---

## C    Supporting Technical Lemmas

**Lemma C.1** (Anytime Hoeffding Inequality [33]). *Let $\{Y_\ell\}_{\ell=1}^{\infty}$ be a martingale difference sequence such that $Y_\ell$ is $Y_\ell \in [a_\ell, b_\ell]$ almost surely for some constants $a_\ell, b_\ell$ almost surely for all $\ell = 1, \cdots, t$. then*

$$\sum_{\ell=1}^{t} Y_\ell \leq 2\sqrt{\sum_{\ell=1}^{t}(b_\ell - a_\ell)^2 \ln\left(\frac{12t^2}{\widetilde{\delta}}\right)}$$

*with probability at least $1 - \widetilde{\delta}$ for all $t \in \mathbb{N}$ simultaneously.*

**Lemma C.2** (Freedman Bounded RVs - unsimplified [19, 8]). *Suppose $\{X_t\}_{t=1}^{\infty}$ is a martingale difference sequence with $|X_t| \leq b$. Let*

$$\mathrm{Var}_\ell(X_\ell) = \mathrm{Var}(X_\ell | X_1, \cdots, X_{\ell-1})$$

*Let $V_t = \sum_{\ell=1}^{t} \mathrm{Var}_\ell(X_\ell)$ be the sum of conditional variances of $X_t$. Then we have that for any $\widetilde{\delta} \in (0, 1)$ and $t \in \mathbb{N}$*

$$\mathbb{P}\left( \sum_{\ell=1}^{t} X_\ell > 2\sqrt{V_t}A_t + 3bA_t^2 \right) \leq \widetilde{\delta}$$

*Where $A_t = \sqrt{2 \ln \ln \left( 2 \left( \max\left(\frac{V_t}{b^2}, 1\right) \right) \right) + \ln \frac{6}{\delta}}$ and $h(s)$ corresponds to horizon index of the state partitions that contains state s.*

**Lemma C.3** (Empirical Bernstein Anytime, Theorem 4 of [29]). *Let $\ell \geq 2$ and $\{Z_t\}_{t=1}^{\ell}$ be i.i.d. random variables with distribution Z satisfying $|Z_t| \leq b$ for all $t \in [\ell]$ Let the sample variance be defined as,*

$$\mathrm{Var}_\ell(Z) = \frac{1}{\ell(\ell-1)} \sum_{1 \leq i < j \leq \ell} (Z_i - Z_j)^2$$

*With probability at least $1 - \delta$,*

$$\mathbb{E}[Z] - \frac{1}{\ell}\sum_{i=1}^{\ell} Z_i \leq \sqrt{\frac{4\mathrm{Var}_\ell(Z)\ln\frac{4\ell^2}{\delta}}{\ell}} + \frac{7b\ln\frac{4\ell^2}{\delta}}{3(\ell-1)}$$

*for all $\ell \in [\mathbb{N}]$.*

**Lemma C.4** (Lemma 7.5 of [2]). *Consider arbitrary $T$ sequence of trajectories $\tau_t = \{s_h^t, a_h^t\}_{h=1}^H$, for $t = 0, 1, \ldots, T$. We have*

$$\sum_{t=1}^{T} \sum_{h=1}^{H} \frac{\mathbf{1}((s_h^t, a_h^t) \in \mathcal{U})}{\sqrt{N_h^t(s_h^t, a_h^t)}} \leq 2\sqrt{|\mathcal{U}| \sum_{(s,a) \in \mathcal{U}} N_{h(s)}^T(s, a)} \leq 2\sqrt{|\mathcal{U}|T}. \tag{19}$$

*Where $\mathcal{U} \subseteq \mathcal{S} \times \mathcal{A}$ is an arbitrary set of state action pairs and $h(s)$ corresponds to horizon index of the state partitions that contains state $s$.*

*Proof.* Consider swapping the order of summation

$$\begin{aligned}
\sum_{t=1}^{T} \sum_{h=1}^{H} \frac{1}{\sqrt{N_h^t(s_h^t, a_h^t)}} &= \sum_{h=1}^{H} \sum_{t=1}^{T} \frac{1}{\sqrt{N_h^t(s_h^t, a_h^t)}} = \sum_{(s,a) \in \mathcal{U}} \sum_{i=1}^{N_{h(s)}^T(s,a)} \frac{1}{\sqrt{i}} \\
&\leq 2 \sum_{(s,a) \in \mathcal{U}} \sqrt{N_{h(s)}^T(s, a)} \leq 2\sqrt{|\mathcal{U}| \sum_{(s,a) \in \mathcal{U}} N_{h(s)}^T(s, a)} \leq 2\sqrt{|\mathcal{U}|T},
\end{aligned} \tag{20}$$

where the first inequality use the fact that $\sum_{i=1}^{N} 1/\sqrt{i} \leq 2\sqrt{N}$ and the second inequality holds due to Cauchy-Schwarz inequality. $\square$

**Lemma C.5** (Sum of inverse counts). *Consider arbitrary $T$ sequence of trajectories $\tau_t = \{s_h^t, a_h^t\}_{h=1}^H$, for $t = 0, 1, \ldots, T$. We have*

$$\sum_{t=1}^{T} \sum_{h=1}^{H} \frac{\mathbf{1}((s_h^t, a_h^t) \in \mathcal{U})}{N_h^t(s_h^t, a_h^t)} \leq 2|\mathcal{U}| \log(T + 1). \tag{21}$$

*Where $\mathcal{U} \subseteq \mathcal{S} \times \mathcal{A}$ is an arbitrary set of state action pairs.*

*Proof.* Consider swapping the order of summation

$$\begin{aligned}
\sum_{t=1}^{T} \sum_{h=1}^{H} \frac{1}{N_h^t(s_h^t, a_h^t)} &= \sum_{h=1}^{H} \sum_{t=1}^{T} \frac{1}{N_h^t(s_h^t, a_h^t)} = \sum_{(s,a) \in \mathcal{U}} \sum_{i=1}^{N_{h(s)}^T(s,a)} \frac{1}{i} \\
&\leq 2 \sum_{(s,a) \in \mathcal{U}} \log\left(N_{h(s)}^T(s, a) + 1\right) \leq 2|\mathcal{U}| \log(T + 1)
\end{aligned} \tag{22}$$

$\square$

# D    Supporting Algorithms

The UCBVI algorithm [8, 2] is described below.

---

**Algorithm 4** UCBVI

---
1: **Input** reward function $r$ (assumed to be known), confidence parameters
2: **for** $t = 1, \ldots, T$
3:     Compute $\widehat{P}_t$ using all previous empirical transition data as $\widehat{\mathbb{P}}_t(s'|s, a) := \frac{N_h^t(s, a, s')}{N_h^t(s, a)}, \forall h, s, a, s'$.
4:     Compute reward bonus $b_h^t(s, a) = 2H\sqrt{\ln(SAHK/\delta)/N_h^t(s, a)}$
5:     Run Value-Iteration on $\{\widehat{\mathbb{P}}_t, r + b_h^t\}_{h=0}^{H-1}$.
6:     Set $\pi_t$ as the returned policy of VI.
7: **End for**

---

UCBVI works by adding a count based bonus to the rewards and then running value iteration over the empirical model based on these bonus augmented rewards. This encourages optimism of the empirical value functions which ensures an appropriate trade-off between exploration and

exploitation. In contrast with previous approaches with provable guarantees for tabular RL problems such as UCRL [6].

As described in [8], the sample complexity of UCBVI is (up to low order terms) $\widetilde{\mathcal{O}}\left(H\sqrt{|\mathcal{S}||\mathcal{A}|}\right)$ with variance aware bonuses and $\widetilde{\mathcal{O}}\left(H^{3/2}\sqrt{|\mathcal{S}||\mathcal{A}|T}\right)$ when the bonus terms are computed using simple count based scores as above. Our results are based on a modification of the rates derived in [2] that are of order $\widetilde{\mathcal{O}}(H^2|\mathcal{S}|\sqrt{|\mathcal{A}|T})$. Despite this base rate having a suboptimal dependence in $H$ and $|\mathcal{S}|$ in contrast with the minimax rate of [8] our results still represent an big improvement w.r.t. the un-shaped rates of [8] when the effective state space is small.

## E   Experimental Details

We conducted experiments building on the UCBVI framework outlined in [8]. Specifically, we used the bonus 1 formulation from the UCBVI Algorithm 4 in [8], with the bonus specifically being $\frac{C}{\sqrt{N(s,a)}}$ as the base UCBVI implementation. Our implementation is based on the open source implementation by Ian Osband https://github.com/iosband/TabulaRL, and open source code is attached.

The implementations of UCBVI-BS replaces this bonus by $\frac{C.\widetilde{V}}{\sqrt{N(s,a)}}$. In our implementation we chose the scaling to be 0.1 with a hyperparameter sweep. The $\widetilde{V}$ was chosen by scaling the optimal value function $V^\star$ by factors chosen between $1+\beta$ and $1-\beta$, with *beta* as $0.2, 0.5, 0.9$ (as described in the experimental section).

The environments themselves are open gridworlds of size 8, a corridor of size $10x10$ and the double corridor has size $10x20$. The reward is 0 everywhere except the goal location. Every experiment is averaged over 3 random seeds.

## F   Intuition for $\text{PseudoSub}_\Delta$ and $\text{PathPseudoSub}_\Delta$

Here we provide some intuition for the concepts of $\text{PseudoSub}_\Delta$ and $\text{PathPseudoSub}_\Delta$ in a deterministic chain environment (actions being left and right with deterministic transitions). The environment has a reward of 1 at state 0 and 0 elsewhere. Discount factor is 0.8 here, and the $\widetilde{V}$ has a sandwich factor $\beta = 0.3$. The true optimal value function is shown at the top of Fig. 8. The reward shaping is shown in the middle row of Fig. 8. The concept of pseudosub is shown via the arrows at $(s,a)$ pairs that lie in $\text{PseudoSub}_\Delta$ for this particular choice of $\widetilde{V}$ and $\Delta$. They basically denote states which are suboptimal with $\Delta$ confidence according to the shaping function $\widetilde{V}$. Now given this definition of $\text{PseudoSub}_\Delta$, the corresponding $\text{PathPseudoSub}_\Delta$ is shown in the last row. These are states that can be eliminated via reward shaping. As we can see, around half of the states can be eliminated this way, making exploration much more directed.

## G   Results for Corrupted $\widetilde{V}$

We evaluate the behavior of UCBVI-Shaped as we use a corrupted version of $\widetilde{V}$ for the shaping in Fig. 9. In particular, we do this by constructing the sandwiched value function $\widetilde{V}$ as described in our experimental evaluation (in this case with $\beta = 1.5$ in the corridor environment) and then adding gaussian noise to corrupt the value function. We experiment with corruptions of 0., 0.1, 0.5, 1.0 variance, zero-centered gaussian noise. The resulting regret is seen in Fig. 9. As expected, the results show that corruption of $\widetilde{V}$ hurts the results as more and more corruption happens, but still performs better than without any shaping at all.

## H   Results for RND

We also ran numerical simulations (in Fig. 10) with a neural network based "pseudo-count" method - random network distillation (RND) [12]. This trains a neural network against a random pre-initialized

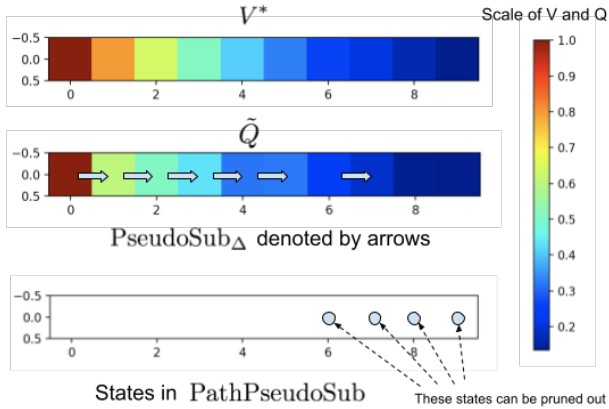

Figure 8: Illustration of the concepts of PseudoSub$_\Delta$ and PathPseudoSub$_\Delta$. Top row shows the optimal value function for this chain MDP (colorbar shows scale). Middle row shows the $\tilde{Q}$ obtained from the shaping and the $(s,a)$ pairs which are in PseudoSub$_\Delta$ (denoted by arrows). From PseudoSub$_\Delta$, the bottom row shows states in PathPseudoSub$_\Delta$ which can be pruned out via shaping.

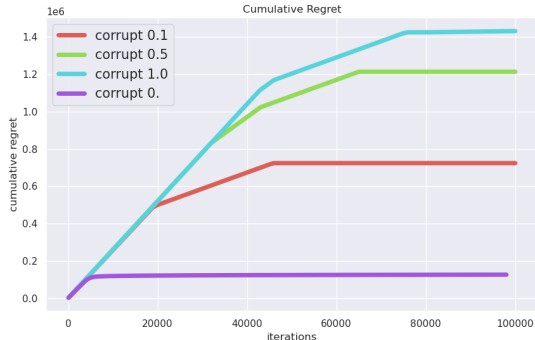

Figure 9: Illustration of behavior with corrupted shaping functions $\widetilde{V}$ with varying levels of corruption

target and uses projection error as a "pseudocount". We ran UCBVI-Shaped-BS with the inverse counts typical in UCBVI-shaped replaced with model error. The results in Fig. 10 are considerably slower than with exact counts likely due to noise in counts estimation, although the results are likely to be improved with environment specific tuning.

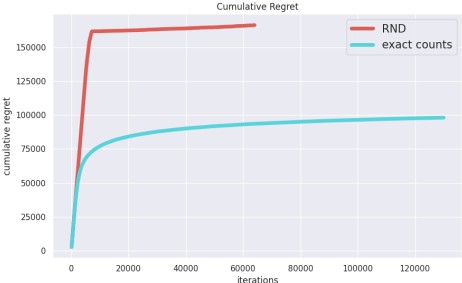

Figure 10: Illustration of behavior of UCBVI-Shaped-BS with RND based pseudocounts instead of exact counts.

# I   Discussion of Applicability of Reward Shaping Assumption

There can't be any way to obtain an improvement to the UCBVI regret bounds by using a reward shaping mechanism unless there is some information about the optimal policy / optimal value function encoded in the shaping function. In other words, there cannot be any free lunch in this setting. Prior knowledge is fundamental for reward shaping to be successful.

In this work we have chosen to make the assumption that the side information available to us $\widetilde{V}$ can be used to sandwich the value of $V^\star$. There are many examples where this assumption may hold.

1. Maze like environments. In any goal metrized goal oriented environment (for example a maze) with a stay-in-place action, if the position of the goal state is (roughly) known and the reward function equals -1 at every step from a state to a neighboring state and 0 for the stay-in-place action at the goal state, using the metric distance from the current state to the goal state may be a good estimator of the number of steps needed to reach the goal $(-V^\star)$. The accuracy of this approximation will depend on how off this heuristic is.

2. Sim to Real. We may think of training a robot in a simulation environment and use the learned optimal value function in the simulation as our input $\widetilde{V}$. Having this information at hand should make training in real much easier than from scratch depending on how accurately the $\widetilde{V}$ estimator from the simulation tracks $V^*$. Under the assumption that dynamics are lipschitz bounded, and there is bounded state estimation error, dynamics estimation error, the resulting value function also shares the sandwich property of the true value function.

3. Object manipulation. In this setting even if the objective is to manipulate a huge combination of objects, an appropriate combination of euclidean distances of objects can serve as a good proxy for $V^\star$. While this may underestimate the distance to the goal especially in the presence of obstacles, in most cases a sandwich term $\beta$ can be found that bounds the true value function (under the assumption that the number of obstacles is bounded). This becomes particularly pronounced as the number of objects increases and there is a combinatorially large state space, all of which cannot possibly be explored.

4. Motion planning: Recent methods have attempted to combine motion planning and RL [18]. One such instantiation would involve using an anytime motion planning to generate a reward function to guide RL. An anytime motion planner may start by generating suboptimal paths, and gets better over time. If the motion planning is run only for some time so as to get a beta-sandwiched suboptimal path (and hence reward shaping as a sandwich on $V^\star$) to guide RL, this would then provide suboptimal shaping that can then be improved with subsequent reinforcement learning as described in our work.

5. First person RPG: For games like first person RPGs, reward shaping terms may look like a sum of euclidean distances to different objects of interest. This will underestimate the true value function will sandwich the optimal value function depending on how many obstacles are in the environment. This environment will also likely benefit from pruning off large parts of the state space.

In all of these situations, it is possible to construct a multiplicative approximation to $V^\star$ to be used as $\widetilde{V}$.

# J   Limitations

A limitation of our work is that the reward shaping has to be provided before hand rather than learned or inferred, which would be very interesting to explore in future work. Additionally, the shaping bound we have is a point-wise one rather than a correlation based bound in expectation, which can be susceptible to be outlier rewards which have a large sandwich term $\beta$. This should be investigated in future work in more detail. Additionally, it is likely that reward shaping can directly allow for horizon reduction in value computation, which should be considered in future work more directly. The behavior of these types of methods under function approximation also may be quite different, which should be studied more systematically.

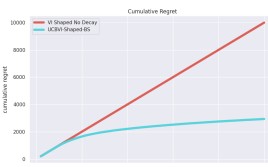 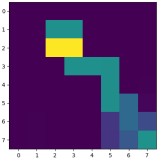 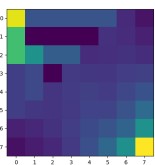

(a) Quantitative comparison of effect of (b) Visualization of non decaying agent get- (c) Visualization of decaying agent
bonus decay ting stuck (UCBVI-Shaped) reaching goal

Figure 11: Effect of bonus decay on the performance of UCBVI-Shaped vs standard reward shaping under shaping misspecification. As we can see, without decay the agent can stuck in arbitrarily sub-optimal points, whereas with decay the agent easily converges to an optimal solution.

## J.1 Does decaying suboptimal reward shaping help over standard shaped rewards?

We ran simulations to understand whether the adaptive decay of $\widetilde{V}$ by $\frac{1}{\sqrt{N(s,a)}}$ actually provides tangible benefits, specifically when the shaping is suboptimal. We compared UCBVI-Shaped with a variant where $\widetilde{V}$ is added to the reward linearly, as done in many practical RL implementations. As seen in Fig. 11, with suboptimal shaping, UCBVI-shaped dampens the effect of the suboptimal shaping and succeed, whereas simple addition of shaping leads to the agent becoming trapped. This suggests that not only can reward shaping improve sample efficiency, but incorporating reward shaping via UCBVI-Shaped can mitigate the potential bias from suboptimal shaping.