# OpenReview forum: "Unpacking Reward Shaping: Understanding the Benefits of Reward Engineering on Sample Complexity"
_NeurIPS.cc/2022/Conference — NeurIPS 2022 Accept_

### Official Review · Reviewer_Xcjp · 2022-07-11

**Rating:** 8
**Confidence:** 4
**Soundness:** 4 excellent
**Presentation:** 4 excellent
**Contribution:** 4 excellent

**Summary:**

Reward shaping is used to speed up reinforcement learning, i.e., it is expected to make exploration more focused and to allow the algorithm to find the optimal policy without the need to explore the entire state space. This means that reward shaping is expected to reduce regret. This is exactly what the authors of this paper study within the family of UCB algorithms. These algorithms use an exploration bonus, which can be seen as a shaping reward. The authors assume that some background knowledge is available. This knowledge can be used to design an informative exploration bonus. This assumption is absolutely correct since this is a common requirement in research on reward shaping, e.g., potential-based reward shaping requires the potential function that can be seen as a very similar assumption. Though I have to admit that the multiplicative bounding of the current paper is a bit strong an assumption, but I think that the contribution is still excellent. The theoretical investigation in this paper is relevant and timely. Having some insights into the impact of reward shaping on regret is very useful, and it is remarkable that the authors managed to get rid of the impact of the horizon H. The two uses of shaping information presented in this paper lead to an interesting algorithm. Empirical evaluation on three domains is also provided, and the experiments confirm the merit of the contribution.

**Questions:**

Initially I wanted to ask how V was determined, but this was clarified in the appendix. I don't have any specific questions.

This paper studies the impact of reward shaping on regret, and it generated insightful results. I feel that it would be interesting if, either the authors of this paper or someone else with similar skills, analysed  the impact of policy or representation transfer [2] or other tweaks with an assumption of optimism [3] that make the exploration more focused. This is of course beyond the scope of this paper, but it would beneficial for the community if this was explored in the future.

[1] Asmuth, John, Michael L. Littman, and Robert Zinkov. "Potential-based Shaping in Model-based Reinforcement Learning." AAAI. 2008.
[2] Taylor, Matthew E., and Peter Stone. "Representation Transfer for Reinforcement Learning." AAAI Fall Symposium: Computational Approaches to Representation Change during Learning and Development. 2007.
[3] Grzes, Marek, and Daniel Kudenko. "PAC-MDP learning with knowledge-based admissible models." Proc. of AAMAS. (2010): 349-358.

Fonts in Figures 3, 5, 6 are too small.

The paper reads well, and I did not see technical problems, but I have to admit that I only scanned the proofs in the appendix. I would probably need to spend a week or more if I wanted to trace all the steps in the proofs.

**Limitations:**

I am not sure if there is anything to say here.

**Strengths And Weaknesses:**

The paper addresses a very relevant and useful question, and it provides a convincing and competent answer.

This is something that can be fixed easily, but the paragraph entitled "Practical Reward Design and Reward Shaping" is a bit unclear because the authors seem to be unaware of one topic in the related literature. The authors indicate that the potential-reward shaping methods try to cover all states and the shaping reward is unknown. This is not entirely true because the literature on potential-based reward shaping has good insights into that, and optimism is also studied and required. For an excellent treatment of this issue, the authors could check [1]. I feel that the current discussion in lines 87-96 is a bit misleading because [1] shows that potential-based reward shaping is a principled method too, the right bias is known (i.e., we know which information is required), and the number of explored states can be reduced (even in the Rmax type of algorithms). As I said above, this is a minor comment, but I hope that the authors could address it in this paragraph.

---

> ### Author Response · Authors · 2022-08-02
> **Response to reviewer comments**
>
> Thank you for your assessment and constructive suggestions!
>
> > it would be interesting if, either the authors of this paper or someone else with similar skills, analysed the impact of policy or representation transfer [2] or other tweaks with an assumption of optimism [3] that make the exploration more focused.
>
> We definitely agree, this seems like an extremely promising direction. Some empirical work has started to explore this direction by learning exploration/curiosity functions across tasks [1] but a deeper theoretical analysis would be an exciting direction to explore next.
>
> >  This is not entirely true because the literature on potential-based reward shaping has good insights into that, and optimism is also studied and required. For an excellent treatment of this issue, the authors could check [1]. I feel that the current discussion in lines 87-96 is a bit misleading because [1] shows that potential-based reward shaping is a principled method too, the right bias is known (i.e., we know which information is required), and the number of explored states can be reduced (even in the Rmax type of algorithms).
>
> Thank you for pointing us to this work, we have added a discussion of this to the text in paragraph “Practical Reward Design and Reward Shaping” of Section 2. [Asmuth et al] indeed shows that potential based reward shaping can be incorporated with methods like Rmax style algorithms, and can help indicate which information is required. However it is non-trivial to compare the analysis with [Asmuth et al] because it does not perform a formal sample complexity analysis (just proving that it is PAC-MDP) and hence it becomes challenging to compare the benefits of potential based vs non potential based shaping from a sample complexity perspective. We think this would be a really exciting direction of future work and would be excited to build toward this!
>
> [1] Meta-learning curiosity algorithms, Ferran Alet, Martin F. Schneider, Tomas Lozano-Perez, Leslie Pack Kaelbling, ICLR 2020

---

### Official Review · Reviewer_PYg1 · 2022-07-11

**Rating:** 5
**Confidence:** 2
**Soundness:** 3 good
**Presentation:** 3 good
**Contribution:** 2 fair

**Summary:**

This paper investigates the regret analysis of a particular form of reward shaping where the shaping function is an approximation of the optimal state value function. This reward shaping is integrated into an extension of the UCBVI algorithm to obtain UCBVI-Shaped. Experiments show benefits of this approach by enabling the elimination of potentially large subsets of the state space by ignoring regions where the value is low during exploration.

**Questions:**

1. Can you justify the assumption that the reward shaping function is an approximation of the optimal state value function? How strong of an approximation does it need to be? If there is a close enough approximation, this trivializes the problem.
2. Why do you choose the bonus term to be the empirical second moment of \tilde{V}?

Minor comments:
* "has focused smarter" -> has focused on smarter
* In Algorithm 2, line 5: should the k superscript be a t? Should \tilde{V} have a subscript of h?
* "\beta . ."
* "\beta = \{1.5, 1.9\}" -> \beta \in \{1.5, 1.5\}



**Limitations:**

The authors state that Section 7 covers the limitations of their work. It does not seem to. But, elsewhere, it is stated that for some experiments (like the single corridor example), the reward shaping may not be very informative and something like an optimistic shaping function can yield better results.

**Strengths And Weaknesses:**

Strengths
* The proposed algorithm has good sample complexity and a detailed regret analysis

Weaknesses
* It seems like a strong assumption is being made whereby the shaping signal is an approximation of the optimal value function.

---

> ### Author Response · Authors · 2022-08-02
> **Response to reviewer comments**
>
> Thank you for your assessment and constructive suggestions!
>
> > Can you justify the assumption that the reward shaping function is an approximation of the optimal state value function? How strong of an approximation does it need to be? If there is a close enough approximation, this trivializes the problem.
>
> We address this concern in the common response, providing a number of intuitive and theoretical explanations for the assumption. We show empirically that even with a significantly loose sandwich assumption, reward shaping still provides significant gains over standard RL and the exact sample complexity bound is shown in Theorem 5.2, showing a dependence on the sandwich term $\beta$ in the leading term.
>
> > Why do you choose the bonus term to be the empirical second moment of \tilde{V}?
>
> This is because of the use of the empirical Bernstein bound in the proof of Lemma A.1 on page 15 in the Appendix. Please refer to this proof for a detailed explanation. This allows us to derive a bound that trades off dependence on $H$ with a value of $\tilde{V}$.
>
> > The authors state that Section 7 covers the limitations of their work. It does not seem to. But, elsewhere, it is stated that for some experiments (like the single corridor example), the reward shaping may not be very informative and something like an optimistic shaping function can yield better results.
>
> Thank you for pointing this out. A limitation of our work is that the reward shaping has to be provided before hand rather than learned or inferred, which would be very interesting to explore in future work. Additionally, the shaping bound we have is a pointwise one rather than a correlation based bound in expectation, which can be susceptible to be outlier rewards which have a large sandwich term $\beta$. This should be investigated in future work in more detail. The point being made in the corridor example is that because there is not a large part of the state space that can be pruned (since there is only one way to go), the benefit is smaller than if we consider the double corridor where much of the state space can be pruned. We have added a discussion of this to Appendix J.
>
> We have also resolved minor comments, thank you for pointing them out!

---

> > ### Comment · Reviewer_PYg1 · 2022-08-06
> > **Thanks!**
> >
> > You have addressed one my concerns on the knowledge required for having an approximate value function to sandwich the true value function. It is not as restrictive as I thought and I have updated my score accordingly.

---

### Official Review · Reviewer_3Wkq · 2022-07-13

**Rating:** 8
**Confidence:** 4
**Soundness:** 4 excellent
**Presentation:** 4 excellent
**Contribution:** 4 excellent

**Summary:**

This paper attempts to shed light on reward shaping (RS) in reinforcement learning (RL) from a theoretical perspective. The authors contend -- correctly, in my opinion -- that the majority of work in RS has focused on RS from a practical point-of-view, while there is relatively less work developing theoretical characterizations of it.

The authors ask the question *"Can we theoretically justify the sample complexity benefits that reward shaping from prior domain knowledge can provide reinforcement learning?"*. In seeking an answer to this question, the authors make some assumptions (e.g., bounds on $\tilde{V}$) and propose a specific algorithm, UCBVI-shaped, which is amenable to analysis.

Given the set-up describe in the preceding paragraph, the contributions of this paper are:
1. A new algorithm that incorporates the RS method of the paper called UCBVI-shaped
2. A theoretical upper-bound on the regret of this method showing lower regret compared to vanilla UCBVI (along with some mathematical tooling, e.g., pseudosuboptimal states)
3. Empirical analysis of the proposed method showing its strengths and weakesses.

In short, the paper puts forth a question and then quite thoroughly studies it. The end result is an affirmation in the sense that the authors are able to derive a bound on regret when they enhance an algorithm (UCBVI) to use their specific instantiation of RS.

**Questions:**

*Suggestions for Improvement*
1. One of my main issues while reading this paper was that the UCBVI algorithm was not described. As a result, some of the points of the text were lost on me as I list below.
* L63 refers to some $H$. I'm not sure where this came from as it isn't previously defined, and so I can only assume it has something to do with UCBVI.
* Moreover, some of the argument in Section 5 compares the regret of UCBVI and UCBVI-shaped. It's difficult to understand this as UCBVI isn't discussed.

2. How does UCBVI-shaped fare if we corrupt $\tilde{V}$? I'm quite curious about this experiment.

3. I'd have been interested in seeing some experiments where UCBVI-shaped uses RND as a method to compute "pesudo-counts"?

4. Figure 4e - Why is there that one state with really high visitations?

5. Figure 3e - What happened to UCBVI-shaped? Some explanation of this peculiar degradation would be helpful

6. L100 "...H is the problem horizon..." - Is H fixed? Can you clarify exactly what H means?

7. Number of seeds used in the experiments is not mentioned?

**Strengths And Weaknesses:**

*Strengths*
1. In terms of quality and scientific soundness, this paper is difficult to fault. The object of inquiry is clearly stated, as are the assumptions made of the investigation into it. A theoretically justified answer to the question is derived, and the paper goes even further by analysing the strengths and weaknesses of the method upon which their theoretical results are based.
2. Moreover, in terms of originality, the paper deserves merit. I can only think of one other work which attempts to develop some theoretical results about RS (which is cited -- Ng et al., 1999).
3. Modulo one major complaint and a few minor issues (described below), the paper is also eminently readable and clear. I rarely found myself repeatedly re-reading anything to try to deduce what was being said.

*Weaknesses*
1. My central issue with the paper is its significance. By this I mean that the assumption of the type of shaped reward function -- $\tilde{V} \leq V^* \leq \beta \tilde{V} $ -- limits the impact and applicability of the results of the paper. It is a significant assumption. In fact, I would argue that it's almost expected that given such a form for the RS that there would be benefits to sample complexity. I know your assumption is looser than this, but the following is just to explain  why I made the last statement. Suppose we have access to the $V^*$. We can essentially just set the values of the $V$ we are learning to the intrinsic rewards generated by $V^*$, and this will almost certainly benefit sample complexity.  While I do appreciate that some simplifying assumptions are required, I think this really limits the scope and impact of this paper. With that said, it's interesting to see how sample complexity is affected by pseudosuboptimal states and so on.
Note: I may have misunderstood/missed something here, in which case I urge the authors to clarify.

2. Further to the point above, I'm also a little unconvinced about the characterization of what kind of problems RS helps. The intuition about dubious paths makes sense in grid-worlds, but I don't see exactly how this intuition generalizes to general RL problems.

*Minor Issues*
Furthermore, below is a list of minor textual errors I was able to catch.

* Figure 1 - Very nice, but what do the colours mean?
* Figure 5 appears before Figure 4 in the text. This was a bit confusing for a moment.
* Assumption 3 - What's $[H]$? I don't think this is defined?
* Theorem 3.1/Lemma 5.1 - What's $\delta$? Not sure if I missed something but as far as I can tell it is not defined.

* L3 "...choice of reward function..." - I found this a bit misleading as normally in RL we just accept the reward function that comes with the environment.
* L47 "...tabula-rasa RL" - I don't know what tabula-rasa means
* L47  "...better direct. exploration..." - typo
* L73 "...aim to learning..." - should it be "learn"
* L111 "...high probability having..." - should it be "has"
* L108-117 - very long run-on sentence was quite confusing to parse
* L183 - "bonus_2"?
* L 296 - "As shown in Fig 5., we see that for environments with open paths..." - But Fig. 5 doesn't show Gridworld (which is the env I assume has open paths)

I am open to reconsidering my evaluation if the authors can convince me of the broader applicability of their results in this paper.

[1] - A. Y. Ng, D. Harada, and S. Russell. Policy invariance under reward transformations: Theory and application to reward shaping. In ICML, volume 99, pages 278–287, 1999.

---

> ### Author Response · Authors · 2022-08-02
> **Response to reviewer comments**
>
> Thank you for your assessment and constructive suggestions!
>
> > My central issue with the paper is its significance.
>
> We have addressed this in the common response providing examples of scenarios where this assumption is likely to be reasonable. Please refer to this response for details.
>
> > Further to the point above, I'm also a little unconvinced about the characterization of what kind of problems RS helps. The intuition about dubious paths makes sense in grid-worlds, but I don't see exactly how this intuition generalizes to general RL problems.
>
> We have added a discussion about this to the common response. This can be useful for tasks in robotic manipulation, robot navigation and even in situations like games. Please refer to the common response for a detailed discussion.
>
> Minor Issues:
> > Figure 1 - Very nice, but what do the colours mean?
>
> Green is lower reward, red is higher reward. We have updated the description for the caption, thanks for pointing this out.
>
> > Figure 5 appears before Figure 4 in the text. This was a bit confusing for a moment.
>
> We have resolved this, thank you for the suggestion.
>
> > Assumption 3 - What's [H]? I don't think this is defined?
>
> This refers to the horizon at which the distribution of states being reached is considered.
>
> > Theorem 3.1/Lemma 5.1 - What's $\delta$? Not sure if I missed something but as far as I can tell it is not defined.
>
> The assertion here holds with likelihood $1-\delta$, as is common in PAC bounds.
>
> > L3 "...choice of reward function..." - I found this a bit misleading as normally in RL we just accept the reward function that comes with the environment.
>
> While the RL formalism often accepts rewards that come with the environment, in most applications someone typically has to design this reward and most practitioners in domains from robotics to recommendation systems have a user design their reward functions.
>
> > L47 "...tabula-rasa RL" - I don't know what tabula-rasa means
>
> Tabula rasa here is referring to learning from scratch
>
> > Typos:
>
> We have resolved these in the updated manuscript
>
> > L 296 - "As shown in Fig 5., we see that for environments with open paths..." - But Fig. 5 doesn't show Gridworld (which is the env I assume has open paths)
>
> This was a typo on our end, we meant to compare Fig 3a and 3d instead, whereas Fig 5 shows the corridor and double corridor environments.
>
> > One of my main issues while reading this paper was that the UCBVI algorithm was not described.
>
> We have added a description of this to Appendix D. It is a direct application of upper confidence bound to the RL setting as described in Azar et al [7].
>
> > L63 refers to some H. I'm not sure where this came from as it isn't previously defined, and so I can only assume it has something to do with UCBVI. Moreover, some of the argument in Section 5 compares the regret of UCBVI and UCBVI-shaped. It's difficult to understand this as UCBVI isn't discussed.
>
> We have added a description of this to Appendix D.
>
> > How does UCBVI-shaped fare if we corrupt \tilde{V}? I'm quite curious about this experiment.
>
> We have added this to Fig 7, Appendix G. In these experiments we corrupted $\tilde{V}$ with different variance gaussian noise sampled independently per state. As expected, we see a significant degradation as corruption increases from no corruption to corruption with standard deviation 1.0.
>
> > I'd have been interested in seeing some experiments where UCBVI-shaped uses RND as a method to compute "pesudo-counts"?
>
> We have added this to Fig 8, Appendix H. We can see that there is definitely degradation from the exact counts, although this performance can likely be improved with some environment specific tuning. The experiments take some time to run so we will update with more iterations in the final version.
>
> > Figure 4e - Why is there that one state with really high visitations?
>
> This is the starting state in the double corridor.
>
> > Figure 3e - What happened to UCBVI-shaped? Some explanation of this peculiar degradation would be helpful
>
> In this scenario the beta term has scaled to be significantly large which means that the shaping is not very informative, and can even be misleading since it might upweight states that shouldn’t be and downweight states that should be. In this situation, UCBVI-shaped can be slower than UCBVI since UCBVI just does state coverage.
>
> > L100 "...H is the problem horizon..." - Is H fixed? Can you clarify exactly what H means?
>
> This is referring to the fact that we are in a finite horizon problem setting where the agent is allowed to operate for a fixed number of time steps in an episode. H is fixed for every environment and the returns are summed up over H time steps from the starting state.
>
> > Number of seeds used in the experiments is not mentioned?
>
> We ran 3 seeds per experiment.

---

> > ### Author Response · Authors · 2022-08-07
> > **Looking forward to further discussions!**
> >
> > Dear Reviewer,
> >
> > Thank you for your time and effort in reviewing our work and the many useful suggestions. We have provided detailed clarification to the questions you raised and provided a set of realistic domains where our assumptions hold, discussed comparisons to related work and corrected many of the errors in the original document. If our response has addressed your concerns, we would be grateful if you could re-evaluate our work.
> >
> > If you have any additional questions or comments, we would be happy to have further discussions

---

> > > ### Author Response · Authors · 2022-08-09
> > > **Follow up**
> > >
> > > Hi Reviewer 3Wkq,
> > >
> > > We understand reviewer load is high and we thank you again for your time!
> > >
> > >
> > > We just wanted to flag that other reviewers shared your concerns about our 'sandwich' assumption on $\tilde V$ and that our response has allayed their concerns to the point of prompting them to raise their score. We are hoping that in case you feel our response addresses your concerns you might consider reassessing the borderline score you have given our submission.
> > >
> > > Thank you!

---

### Official Review · Reviewer_utjg · 2022-07-18

**Rating:** 7
**Confidence:** 4
**Soundness:** 3 good
**Presentation:** 2 fair
**Contribution:** 3 good

**Summary:**

Learning near-optimal policies in sparse reward environments with large state space is challenging. In theoretical reinforcement learning (RL) literature, it has been understood that directed (optimistic) exploration strategies (provably) improve sample complexity over random exploration techniques. However, in practice, people use heuristic-based (exploiting domain knowledge) reward shaping to demonstrate successful results. This paper conducts a formal theoretical investigation on the impact of such reward shaping methods on the sample complexity of RL algorithms. In particular, the authors propose two simple modifications to the well-known UCBVI algorithm and show how these changes lead to tight regret bounds. In the experiments, they studied:
- the importance of the two modifications that they introduced,
- the type of environments where the utility of reward shaping can be realized, and
- the impact of the quality of the shaping function on the sample complexity.

**Questions:**

Questions:

1/ I couldn't fully understand Assumption 3; please explain with an example and point me where it is used in your analysis.

2/ I have noticed several inconsistencies in the theoretical section, for example:
- In Eq.(1), $N_h^t$ is already defined, but $N_t$ is not.
- Does $\widehat{P}_t$ depend on $h$? In line 168, it is defined based on $N_h^t$ not $N_t$.
- $\widetilde{V}$ is a function of $\mathcal{S}$ only not a function of $\mathcal{S} \times \mathcal{A}$, right? In Assumption 1, line 174, and Algorithm 2, $\widetilde{V}(s,a)$ is used.

Please fix them.

3/ The PseudoSub and PathPseudoSub sets introduced in Section 5.1 are clearly explained. However, illustrating these notions for some specific environments (e.g., deterministic chain MDP) would help the reader understand the tightness/improvement of the regret bound presented in Theorem 5.2.

4/ Regarding section 6.2, it would be interesting to consider a formal investigation on the type of environments where the reward shaping function improves/hurts the learning process. See Section 5 of [Dann et al. 2022].

5/ Regarding section 6.3, how does your work theoretically/empirically relate to [Cheng et al. 2021], who also investigate the quality of shaping functions in improving the sample complexity.

References:

[Dann et al. 2022] Dann et al. Guarantees for Epsilon-Greedy Reinforcement Learning with Function Approximation. ICML, 2022.

[Cheng et al. 2021] Cheng et al. Heuristic-Guided Reinforcement Learning. NeurIPS, 2021.



**Limitations:**

Their proposed method does not seem to cause any direct potential negative societal impacts.

**Strengths And Weaknesses:**

**Post Rebuttal:** I thank the authors for the detailed response that addresses most of my concerns. I hope that the authors can incorporate the details in the rebuttal in the updated version of their draft. I have updated my score accordingly.

----

Strengths:

Despite the practical success of reward shaping (exploiting domain knowledge), theoretical understanding (specifically in sample complexity) is still lacking in the literature. Thus, the formal investigation of this paper is an important step in this direction. The proposed changes to the UCBVI algorithm are theoretically motivated, novel, and easy to implement.

The related work is clearly discussed in Section 2, and the current work is clearly positioned in the literature.

****

I have checked the theoretical claims in the main paper but haven't checked the proofs (or verified the correctness) in the appendix. However, the theoretical claims seem to match with the intuition. The authors have explained the role of reward shaping in pruning the unnecessary part of the state space in the theoretical section.

Since the paper is theoretical, the conducted experiments are sufficient/reasonable enough to flesh out the essential characteristics of reward shaping (or to validate the paper's central claims).

****

Weaknesses:

For most of the part, the paper is well written; the motivation and the story are well conveyed. However, the writing of the theoretical section can be improved, and some intuitive examples would also help non-theoretical readers.

Also, see the questions below.

---

> ### Author Response · Authors · 2022-08-02
> **Response to reviewer comments (1/2)**
>
> Thank you for your assessment and constructive suggestions!
>
> > some intuitive examples would also help non-theoretical readers.
>
> We have added some intuitive examples from robotic navigation and manipulation that provide geometric intuition beyond just the theoretical results in the common response and in Appendix I. Please refer to the common response for a more detailed discussion.
>
> > I couldn't fully understand Assumption 3; please explain with an example and point me where it is used in your analysis.
>
> This assumption is there for simplicity, this allows us to consider counts of the form $N_h^t$ (indexed by $h$) instead of global counts $N^t$. This assumption is simply saying the states are time indexed. That is the states reachable at step h are disjoint from those reachable at step h’. It is a commonly used assumption when analyzing tabular RL algorithms that simplifies some of the calculations and the presentation of the results.
>
> >I have noticed several inconsistencies in the theoretical section, for example:
> In Eq.(1), $N_h^t$ is already defined, but $N_t$ is not.
> Does $\hat{P}_t$ depend on $h$? In line 168, it is defined based on Nht not Nt.
> $\tilde{V}$ is a function of S only not a function of S×A, right? In Assumption 1, line 174, and Algorithm 2, $\tilde{V}(s,a)$ is used.
>
> Thanks for pointing these parts out! In equation (1), the N_t is a typo, it should be N_h^t instead of N_t, we apologize for the confusion and the typo. (2) Thanks for pointing this out, your understanding is correct. The $\hat{P}_t$ are horizon dependent. (3) We apologize for the typo again, they both should be $\tilde{V}(s)$. These are resolved in the text.
>
> > The PseudoSub and PathPseudoSub sets introduced in Section 5.1 are clearly explained. However, illustrating these notions for some specific environments (e.g., deterministic chain MDP) would help
>
> We have added an illustration of this in a deterministic chain MDP in Appendix F, Fig 6. The top row shows $V^*$, the next row shows the (s,a) pairs in delta-pseudosub for some imperfect reward shaping with $\beta = 0.3$, and $\delta = 0.1$. Intuitively they are the state-action pairs which can actually be ruled out using the noisy reward function. Given these, PathPseudoSub is illustrated in the third row, which involves all the states which cannot be reached without traversing a (s, a) pair in delta-pseudosub. As is clear from the figure, these states are states “away” from the goal when starting from the middle of the chain and hence can be eliminated without having to actually explore them. Standard reinforcement learning or UCBVI would give equal weight to exploring these states as well, which is where the benefits come from.
>
> > Regarding section 6.2, it would be interesting to consider a formal investigation on the type of environments where the reward shaping function improves/hurts the learning process. See Section 5 of [Dann et al. 2022].
>
> Thank you very much for bringing this up. Indeed this work is related in so far as it is part of a growing literature on instance dependent complexity measures for reinforcement learning. In their case they propose a measure of complexity that quantifies when is epsilon greedy a good exploration strategy. Our work is related to this goal in that we also propose measures of complexity under which reward shaping is useful, in our case delta-pseudosub and delta-path-pseudosub. These measures are connected to the sample complexity through Theorem 5.2, giving us a metric for how we’d expect shaping to help in the learning process in terms of sample complexity. Let’s go over the reviewer questions one by one. 1) When will shaping hurt the learning process? Under our assumptions, our results reduce to the original UCBVI bounds when the value of $\beta \rightarrow \infty$. Thus, as long as the sandwich condition for $\tilde V$ holds, shaping does not affect the regret rates of UCBVI and can only make them smaller. Thus, shaping can only be beneficial as long as the sandwich condition holds. In the new experiments we have added in the rebuttal version we explore a setting where $\Vtilde$ has been corrupted by random noise and therefore the sandwich assumption may not hold. We see that in these cases shaping is still useful. It would be interesting to fully characterize when is $\tilde V$ is useful in the case it doesn’t fully satisfy the sandwich condition. 2) When will shaping help a lot in the learning process? As we have explained in the shared response, we expect shaping to be beneficial when the MDP has certain connectedness structure as characterized by the maze and disjoint room example. This can be easily inferred from the definitions of the pseudosub, boundarypseudosub and pathpseudosub sets. We think a deeper formalization of these in terms of the MDP dynamics would be a fantastic avenue for future work!

---

> > ### Author Response · Authors · 2022-08-02
> > **Response to reviewer comments (2/2)**
> >
> >
> > > Regarding section 6.3, how does your work theoretically/empirically relate to [Cheng et al. 2021], who also investigate the quality of shaping functions in improving the sample complexity.
> >
> > [Cheng et al. 2021] studies how reshaping the reward $r$ by using a heuristic function $h$ affects sample efficiency. In particular, when equipped with a heuristic they consider a “reshaped” MDP with a lowered discount factor and a reward modified with the heuristic using a mixing factor. This allows them to lower variance since the lowered discount factor provides lower variance estimates, allowing them to perform “short-horizon” reasoning using the heuristic. This reshaped MDP is reduces back to the original MDP at convergence. This allows for a favorable bias-variance tradeoff during learning, using the heuristic to ensure that short horizon reasoning biases exploration in meaningful directions.
> >
> > While both works do use the shaping to improve sample efficiency, the gain comes in largely complementary ways. The gain of this work largely comes from horizon reduction (gaining in the |H| factor) using the shaping, in our work the gain largely comes from pruning the state space to avoid exploring unnecessary parts of the state space (gaining on the |S| factor through |S\PathPseudoSub_Delta|) and sharpening the range of values that the estimated values can take ($\beta \tilde{V}$). In future work, bringing these ideas together would allow us to obtain gains both in terms of state space pruning and in terms of shortening the horizon of value estimation.
> >
> > Additionally, since the mechanisms for incorporating of the value function are different ([Cheng et al] do it through lowering discount and modifying reward with a linear additive term) whereas our work does through bonus scaling and value projection, the properties required of the resulting heuristics (or shaping functions) are different as well. While both works will benefit from the shaping being the optimal value function, our work performs better with better “sandwiched” value functions which increase the size of PathPseudoSub-Delta, while [Cheng et al] benefit from pessimistic heuristics where greedy (or short horizon) reasoning provides optimal policies.
> >
> > Also, [Cheng et al. 2021] mainly focuses on providing a population version of regret analysis and characterizing the bias-variance tradeoff from learning in the “reshaped” MDP,  rather than providing a non-asymptotic sample complexity bound. Our results provide a non-asymptotic sample complexity bound (with respected to the confidence parameter $\beta$ and PathPseudoSub-Delta). Overall, we think the strengths of these papers are largely complementary and we would be keen to investigate combinations in the future.

---

> > > ### Author Response · Authors · 2022-08-07
> > > **Looking forward to further discussions!**
> > >
> > > Dear Reviewer,
> > >
> > > Thank you for your time and effort in reviewing our work and the many useful suggestions. We have provided detailed clarification to the questions you raised, discussed comparisons to related work and corrected many of the errors in the original document. If our response has addressed your concerns, we would be grateful if you could re-evaluate our work.
> > >
> > > If you have any additional questions or comments, we would be happy to have further discussions

---

> > > > ### Comment · Reviewer_utjg · 2022-08-07
> > > > **Thanks for the response**
> > > >
> > > > I thank the authors for the detailed response that addresses most of my concerns. I hope that the authors can incorporate the details in the rebuttal in the updated version of their draft. I will update my score accordingly.

---

> > > > > ### Author Response · Authors · 2022-08-09
> > > > > **Follow up**
> > > > >
> > > > > Thanks so much for your answer. We will incorporate all these into our final manuscript for which we'll have an extra page.

---

### Author Response · Authors · 2022-08-02
**Shared Response to All Reviewers (1/2)**

We thank the reviewers for their thoughtful comments and responses. We first address some questions that were brought up by several of the reviewers:

> When is the assumption about V_tilde reasonable?

We would like to mention that as Reviewer Xcjp has correctly pointed out, there can’t be any way to obtain an improvement to the UCBVI regret bounds by using a reward shaping mechanism unless there is some information about the optimal policy / optimal value function encoded in the shaping function. In other words, there cannot be any free lunch in this setting. Prior knowledge is fundamental for reward shaping to be successful.

In our work we have chosen to make the assumption that the side information available to us $\tilde{V}$ can be used to sandwich the value of $V^*$. There are many examples where this assumption is likely to hold.

1. Maze like environments. In any goal metrized goal oriented environment (for example a maze) with a stay-in-place action, if the position of the goal state is (roughly) known and the reward function equals -1 at every step from a state to a neighboring state and 0 for the stay-in-place action at the goal state, using the metric distance from the current state to the goal state may be a good estimator of the number of steps needed to reach the goal ($-V^*$). The accuracy of this approximation will depend on how off this heuristic is.

2. Sim to Real. We may think of training a robot in a simulation environment and use the learned optimal value function in the simulation as our input $\tilde V$. Having this information at hand should make training in real much easier than from scratch depending on how accurately the $\tilde{V}$ estimator from the simulation tracks $V^*$. Under the assumption that dynamics are lipschitz bounded, and there is bounded state estimation error, dynamics estimation error, the resulting value function also shares the sandwich property of the true value function.

3. Object manipulation. In this setting even if the objective is to manipulate a huge combination of objects, an appropriate combination of euclidean distances of objects can serve as a good proxy for $V^*$. While this may underestimate the distance to the goal especially in the presence of obstacles, in most cases a sandwich term $\beta$ can be found that bounds the true value function (under the assumption that the number of obstacles is bounded). This becomes particularly pronounced as the number of objects increases and there is a combinatorially large state space, all of which cannot possibly be explored.

4. Motion planning: Recent methods have attempted to combine motion planning and RL [1]. One such instantation would involve using an anytime motion planning to generate a reward function to guide RL. An anytime motion planner may start by generating suboptimal paths, and gets better over time. If the motion planning is run only for some time so as to get a beta-sandwiched suboptimal path (and hence reward shaping as a sandwich on $V^*$) to guide RL, this would then provide suboptimal shaping that can then be improved with subsequent reinforcement learning as described in our work.

5. First person RPG: For games like first person RPGs, reward shaping terms may look like a sum of euclidean distances to different objects of interest. This will underestimate the true value function will sandwich the optimal value function depending on how many obstacles are in the environment. This environment will also likely benefit from pruning off large parts of the state space.

In all of these situations, it is possible to construct a multiplicative approximation to $V^*$ to be used as $\tilde{V}$. We are very excited to see the insights of our work being applied to all these settings (and more) in future research.

[1] PRM-RL: Long-range Robotic Navigation Tasks by Combining Reinforcement Learning and Sampling-based Planning, Aleksandra Faust, Oscar Ramirez, Marek Fiser, Kenneth Oslund, Anthony Francis, James Davidson, Lydia Tapia, ICRA 2018

---

> ### Author Response · Authors · 2022-08-02
> **Shared Response to All Reviewers (2/2)**
>
> > What types of problems will the reward shaping actually be helpful for?
>
> In our work we have shown that reward shaping may provide two types of benefits, first an improvement in the horizon dependence of the problem and second a better dependence on the size of the effective state space. In the first the savings in the number of samples vs running UCBVI without shaping will scale by a multiplier scaling as $poly(H \tilde{V})$. When the scale of $\tilde{V}$ is much smaller than $H$ this can lead to a substantial speedup.
>
> The most relevant type of speedup we can expect from any reward shaping procedure will come from the improved dependence on the size of the state space. Reward shaping will be useful in problems where $\tilde{V}$ and therefore $\tilde{Q}$ successfully encodes large regions of the state space to be ignored during exploration. In general this is likely to occur in MDPs composed of disjoint regions each of which can be accessed by traversing a small set of states, and where we have access to a $\tilde{V}$ that can help prune out some of these connected components. This is the structure of problems such as mazes, video games (a character choosing what rooms / branches of the decision tree to take) and many navigation tasks. Shaping is also useful in MDPs where states have low local connectivity (this is the case of the vast majority of use cases of RL where teleportation from one state to another far away one is not possible). In these cases shaping can help the agent to explore only within a ‘safe tubular’ section of the state space instead of wandering off parts of the state space that are suboptimal. This can arise in cases such as the sim to real scenario we have described before. After training a robot in simulation a shaping $\tilde{V}$ can help encoding what kind of joint positions a robot should avoid during exploration in the real environment, allowing it to prevent entering catastrophic and unnecessary exploration modes such as bumping into walls or falling over. This can have beneficial consequences not only in the sample efficiency in the real environment but also in making sure exploration is safe.

---

> > ### Comment · Reviewer_Xcjp · 2022-08-04
> > **lets's discuss**
> >
> > The authors can participate in our discussions only until Aug 9th. I thought that it would useful to interact with them.
> >
> > After reading all the reviews and authors' comments, I can see that (in addition to many other useful comments) there were two major concerns mentioned by the reviewers. First, I can see the need for a general explanation as to why it is fair to assume heuristic knowledge has to be in a agreement with the policy. I feel that the authors explained this well, and I would just add this is a very standard assumption in more traditional AI. The classic textbook by Russel & Norvig makes this clear. The second concern is a bit more subtle. It was nicely stated by rev 3Wkq in the their Weakness 1, and also indicated by other reviewers. It is about $\tilde{V}\leq V^*\leq\beta\tilde{V}$. I actually predicted these questions in my initial review, and the abstract of my review comments on both questions (please have a look).
> >
> > The only thing that I could add now is that even though the equation that I copied above makes a strong assumption (even considering the standard AI guidance in Russel & Norvig), I still believe that this paper makes an important step towards better understanding of regret when reward shaping is used. Mathematical theories often have assumptions, and I feel that this the right time to see this work published because it would an important milestone for the community. These authors uncover an important challenge, and provide a competent theoretical solution to a part of it.

---

### Meta-Review · Area_Chair_faUA · 2022-08-26

**Recommendation:** Accept
**Confidence:** Certain

**Metareview:**

The reviewers carefully analyzed this work and agreed that the topics investigated in this paper are important and relevant to the field. Overall, the reviewers had a generally positive impression of this paper. One reviewer argued that this paper addresses a relevant and valuable question and makes an important step towards a better understanding of regret when reward shaping is used. Even though this paper makes assumptions that were of some concern to other reviewers, this reviewer argued that the paper is nonetheless an important milestone for the community. Another reviewer acknowledged that this paper conducted a formal theoretical investigation of the impact of reward shaping methods on the sample complexity of RL algorithms and argued that all proofs seem to be sound. This reviewer had a few technical questions, which were all addressed by the authors. Post-rebuttal, the reviewer encouraged the authors to incorporate the corresponding details (such as those discussed in the rebuttal) in the updated version of their draft. A third reviewer emphasized that this paper shed light on reward shaping from a theoretical perspective. They argued that the quality and scientific soundness of the paper are objectively excellent, that the paper is original, and that it deserves merit. The reviewer pointed out one main weakness, however, regarding the assumption of the type of the shaped reward function. They wondered whether this assumption could limit the impact and applicability of the paper's results. After reading the authors' thorough rebuttal, however, the reviewer stated that they were satisfied with all responses and updated their score accordingly. Finally, a fourth reviewer also had an overall positive view of this work but pointed out, as a weakness, the seemingly strong assumption that the shaping signal is an approximation of the optimal value function. After reading the authors' response, however, the reviewer stated that the assumptions made in this paper were not as restrictive as they initially thought, and updated their score. Overall, thus, it seems like most reviewers were positively impressed with the quality of this work. They look forward to an updated paper version addressing the suggestions mentioned in their reviews and during the discussion phase.

**Award:**

No

---

### Decision · Program_Chairs · 2022-09-14

Accept